# LEARNING PDE SOLUTION OPERATOR FOR CONTINUOUS MODELING OF TIME-SERIES

## ABSTRACT

Learning underlying dynamics from data is important and challenging in many real-world scenarios. Incorporating differential equations (DEs) to design continuous networks has drawn much attention recently, the most prominent of which is Neural ODE. Most prior works make specific assumptions on the type of DEs or restrict them to first or second-order DEs, making the model specialized for certain problems. Furthermore, due to the use of numerical integration, they suffer from computational expensiveness and numerical instability. Building upon recent Fourier neural operator (FNO), this work proposes a partial differential equation (PDE) based framework which improves the dynamics modeling capability and circumvents the need for costly numerical integration. FNO is hard to be directly applied to real applications because it is mainly confined to physical PDE problems. To fill this void, we propose a continuous-in-time FNO to deal with irregularly-sampled time series and provide a theoretical result demonstrating its universality. Moreover, we reveal an intrinsic property of PDEs that increases the stability of the model. Several numerical evidence shows that our method represents a broader range of problems, including synthetic, image classification, and irregular time-series. Our framework opens up a new way for a continuous representation of neural networks that can be readily adopted for real-world applications.

## 1 INTRODUCTION

The modeling of time-series data plays an important role in various applications in our everyday lives including climate forecasting (Schneider, 2001; Mudelsee, 2019), medical sciences (Stoffer & Ombao, 2012; Jensen et al., 2014), and finance (Chatigny et al., 2020; Andersen et al., 2005). Numerous deep learning architectures (Connor et al., 1994; Hochreiter & Schmidhuber, 1997; Cho et al., 2014) have been developed to learn sequential patterns from diverse time-series datasets. In recent years, leveraging differential equations (DEs) to design continuous networks has attracted increasing attention, first sparked by neural ordinary differential equations (ODEs) (Chen et al., 2018). Differential equations that characterize the rates of change and interaction of continuously varying quantities have become indispensable mathematical language to describe time-evolving real-world phenomena (Cannon & Dostrovsky, 2012; Sundén & Fu, 2016; Black & Scholes, 2019). By virtue of their ability to represent and predict the world around us, incorporating differential equations into neural networks has reinvigorated research in continuous deep learning, offering new theoretical perspectives on neural networks. Moreover, they provide memory efficiency, invertibility, and the ability to handle irregular time-series (Rubanova et al., 2019; Chen et al., 2019; Dong et al., 2020).

Despite their eminent success, Neural ODEs have yet to be successfully applied to complex and large-scale tasks due to the limitation of expressiveness of ODEs. To respond to this limitation, there are several works that enhance the expressiveness of Neural ODEs (Gholami et al., 2019; Gu et al., 2021). Another line of works attempts to introduce more diverse differential equations, such as controlled differential equations (Kidger et al., 2020), delay differential equations (Zhu et al., 2020; Anumasa & PK, 2021), and integro-differential equations (Zappala et al., 2022). In real applications, however, we usually know little about the underlying dynamics of the time evolution system. In general, we are hard to knowledge about how the temporal states evolve, which kind of differential equation it follows, how variables depend on each other, and how high derivatives it contains. Therefore, it is necessary to develop a model that can learn an extended class of differential equations that is able to cover more diverse applications, in a data-driven manner (Holt et al., 2022).

In this work, we propose a partial differential equation (PDE) based novel framework that can learn a broad range of dynamics without prior knowledge of governing equations. PDEs that enjoy relations between the various partial derivatives of multivariable states represent much general dynamics, including ODEs as a special case. There have been several attempts to design neural networks through the lens of PDEs (Eliasof et al., 2021; Ruthotto & Haber, 2020; Ben-Yair et al., 2021; Sun et al., 2020; Kim et al., 2020b). Most of the prior works have been designed under specific assumptions about the type or structure of the PDEs. As the underlying dynamics are unknown in real-world data, however, it should be oblivious to the knowledge of the underlying PDE structure and needs to be learned from data. Moreover, because the appropriate properties of PDE differ for each given problem (Eliasof et al., 2021), it is necessary to be able to represent a wide range of PDEs as possible. To this end, we adopt Fourier neural operator (FNO) (Li et al., 2021), an emerging model that directly parametrizes the PDE solution operator without prior information on the governing PDE. By learning the solution operator, the model automatically learns diverse equations in a completely data-driven way, as well as attains the solution in a single call without using numerical integration that invokes computational expensiveness and numerical instability.

Because FNO has been scrutinized on mathematically well-posed PDE problems, adapting FNO to practical applications confronts several drawbacks. First, due to its notion of discrete time, FNO is difficult to directly transfer to irregularly-sampled time-series commonly arising in real-world problems. To render it more suitable for continuous time-series, we propose a continuous-in-time FNO, termed *CTFNO*, that can be evaluated at arbitrary time points. By learning continuous-time solution operator, CTFNO can flexibly capture diverse time-series data. Moreover, we demonstrate the representational power of CTFNO via a rigorous theoretical proof of the universal approximation theorem. Secondly, we develop a network architecture that guarantees stability and leads to well-posed learning problems. By ensuring the stability of the model, it can defend against noisy observations and alleviate overfitting. We also verify that it enhances the adversarial robustness for image classification. The results of various experiments show that our model provides superior performance on wide array of real-world data with applications in time-series and image classification.

## 2 BACKGROUND

**Fourier Neural Operator** Let $\Omega \subset \mathbb{R}^n$ be a bounded domain, and $\mathcal{A} = \mathcal{A}\left(\Omega; \mathbb{R}^{d_a}\right)$ and $\mathcal{U} = \mathcal{U}\left(\Omega; \mathbb{R}^{d_u}\right)$ be spaces of functions from $\Omega$ to $\mathbb{R}^{d_a}$ and $\mathbb{R}^{d_u}$, respectively. Fourier neural operator (FNO) (Li et al., 2021) learns a map $\mathcal{G} : \mathcal{A} \to \mathcal{U}$ between infinite-dimensional function spaces, with a special focus on PDEs. An input function $a \in \mathcal{A}$ is any of source or initial functions, and $u \in \mathcal{U}$ is the corresponding solution. The solution to fairly general PDEs is represented as a convolution operator with kernel $G : \mathbb{R}^n \to \mathbb{R}^{d_u \times d_a}$ called by a Green's function as follows:

$$u(x) = \mathcal{G}(a)(x) = \int_{\Omega} G(x - y) a(y) \, dy, \; \forall x \in \Omega. \tag{1}$$

This solution formula elucidates an elegant way to design FNO. The overall computational flow of FNO for approximating the convolution operator (1) is given as $a \xrightarrow{\mathcal{P}} v_0 \xrightarrow{\mathcal{L}_1} v_1 \xrightarrow{\mathcal{L}_2} \cdots \xrightarrow{\mathcal{L}_L} v_L \xrightarrow{\mathcal{Q}} u$, for a given depth $L$. To recover the sorts of universality, the input function $a$ concatenated with position $x$, denoted by $[a(x); x] \in \mathbb{R}^{(d_a+n)}$, is lifted to a higher dimensional representation $v_0(x) \in \mathbb{R}^{d_v}$ by $\mathcal{P}(a)(x) := P[a(x); x]$ with a matrix $P \in \mathbb{R}^{d_v \times (d_a+n)}$. $\mathcal{Q}$ is a projection operator of the form $\mathcal{Q}(v)(x) := Qv(x)$ for $Q \in \mathbb{R}^{d_u \times d_v}$. Fourier layers $\mathcal{L}_\ell$ are defined as follows:

**Definition 2.1.** *(**Fourier layers** (Li et al., 2021)) For a convolution operator $\mathcal{K}_\ell : \mathcal{U}\left(\Omega; \mathbb{R}^{d_v}\right) \to \mathcal{U}\left(\Omega; \mathbb{R}^{d_v}\right)$ with kernel $\kappa_\ell : \mathbb{R}^n \to \mathbb{R}^{d_v \times d_v}$, a linear transform $W_\ell : \mathbb{R}^{d_v} \to \mathbb{R}^{d_v}$, $R_\ell = \mathcal{F}(\kappa_\ell) : \mathbb{R}^n \to \mathbb{C}^{d_v \times d_v}$, Fourier transform $\mathcal{F}$, and an activation function $\sigma : \mathbb{R} \to \mathbb{R}$ which is component-wisely applied, the $\ell$-th Fourier layer $\mathcal{L}_\ell$ is defined as follows: $\forall v \in \mathcal{U}\left(\Omega; \mathbb{R}^{d_v}\right)$, $x \in \Omega$,*

$$\mathcal{L}_\ell(v)(x) := \sigma\left(W_\ell v(x) + \mathcal{K}_\ell(v)(x)\right) = \sigma\left(W_\ell v(x) + \mathcal{F}^{-1}\left(R_\ell \cdot (\mathcal{F}v)\right)(x)\right),$$

*where $R_\ell \cdot \mathcal{F}(v) : \xi \mapsto R_\ell(\xi)(\mathcal{F}v)(\xi)$ for $\xi = (\xi_1, \cdots, \xi_n) \in \mathbb{R}^n$.*
Both Fourier transform $\mathcal{F}$ and inverse Fourier transform $\mathcal{F}^{-1}$ are implemented by the fast Fourier transform (Nussbaumer, 1981) with truncated frequencies at maximum modes $\max |\xi_i| \leq N$.

**Treatment of time-varying problems** When applied to time-dependent PDEs, the original FNO can only learn an operator that maps the initial function to a solution for a single fixed time. To

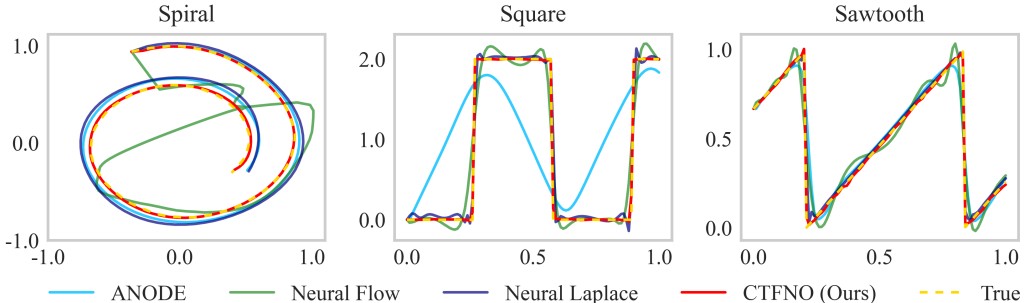

Figure 1: Visualization of learned solutions to synthetic datasets.

deal with time-varying problems, two methods are suggested in (Li et al., 2021): (i) It poses the time-dependent problem as a sequence-to-sequence task. (ii) Treats it as $(n + 1)-$dimensional problem by adding one more dimension and apply FNO layers to convolve in space-time domain. In this case, FNO can only predict the solution at times on a fixed equispaced temporal mesh.

## 3 METHOD

### 3.1 MOTIVATION

**Why do we consider PDEs for time-series modeling?** Starting with Neural ODEs, leveraging DEs has been found to be effective in modeling time-series data (Kidger et al., 2020; Jia & Benson, 2019). They have shown promising results, however, their model architectures and inference schemes are specialized to the DEs on which they are based. These bespoke model structures rule out their generalization ability to other classes of DEs, which is further exacerbated in real-world applications. Therefore, we need a model that can learn a broad range of dynamics to cope with a wide variety of applications. In this paper, we propose a novel framework for modeling time-series data through the lens of PDEs, a richer family of differential equations. Unlike ODEs which deal with functions of a single variable, time-dependent PDEs describe the evolution of a physical quantity $u(t, x)$, not only with time $t$, but also according to a spatial variable $x = (x_1, \cdots, x_n) \in \Omega$, having the form:

$$\frac{\partial u}{\partial t}(t, x) = \mathcal{L}u(t, x) \text{ in } [0, \infty) \times \Omega, \ \ u(0, x) = a(x) \text{ on } \Omega, \tag{2}$$

where $\partial/\partial t$ denotes a partial derivative w.r.t. time $t$, $\mathcal{L}$ is some differential operator consists of spatial derivatives, and $u : [0, \infty) \times \Omega \to \mathbb{R}^{d_u}$ satisfying (2) is called by the solution with initial condition $a(x) \in \mathcal{A}$ at $t = 0$. For example, (2) with $\mathcal{L} = \partial^2/\partial x_1^2 + \cdots + \partial^2/\partial x_n^2$, called by heat equation, describes the diffusion process of temperature on a surface. We refer the interested reader to (Evans, 2010) for more details in PDEs. Due to their heavy expressivity, PDEs are widely used to describe complex continuous processes including the laws of physics (Temam, 2001), principles of engineering (Kulov & Gordeev, 2014), and medical science (Joshi, 2002). Experimental results in Figure 1 conducted on three synthetic examples proposed by (Biloš et al., 2021) confirm the ability of our PDE-based model to represent more general temporal dynamics. It can be seen that our model, which is based on PDEs, which represent a more diverse classes of dynamics than DEs on which baselines are built, consistently achieves superior performance in all examples. Please consult Section 4.1 for additional results and experimental details. In this paper, we harness FNO for modeling a variety of time-series applications through PDEs.

### 3.2 CONTINUOUS-TIME FNO

In a wide variety of real-world problems, the system of interest evolves continuously over time. FNO has recently shown the promise ability to learn complex PDEs, however, the notion of discrete time of FNO hinders its wider applicability to time-varying systems. To ameliorate this limitation, we introduce a *continuous-in-time Fourier neural operator (CTFNO)*. Consider a time-dependent PDE (2). The Green's function formula says that there exists a Green's function $G : [0, \infty) \times \mathbb{R}^n \to \mathbb{R}^{d_u \times d_a}$ such that the solution for (2) is represented by as follows:

$$u(t, x) = \int_\Omega G(t, x - y) a(y) \, dy, \ \forall (t, x) \in [0, \infty) \times \Omega. \tag{3}$$

This shows that the weights of an FNO layer should be conditioned on time, to learn an operator $(t, a(x)) \mapsto u(t, x)$ for an arbitrary time $t$ and initial condition $a$. For example, the Fourier transform

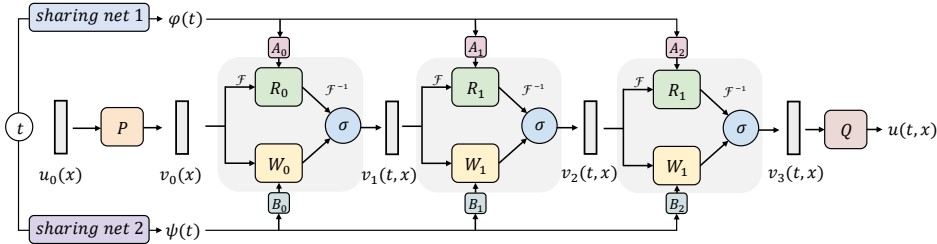

Figure 2: The visualization of CTFNO architecture. Two time embedding networks specify the time t into hidden representations $\varphi(t)$ and $\psi(t)$. Then $\varphi(t)$ and $\psi(t)$ incorporate the temporal information into each Fourier layer through time modulation operators $A_\ell$ and $B_\ell$. See Appendix C.2 for details.

of the Green's function of the heat equation is given by $R(t, \xi) = e^{-4t\xi^2}$. To this end, we propose a time-aware Fourier layer that updates the solution based on (3) as follows:

**Definition 3.1.** *(Continuous-time Fourier layers) Under the same notations with Definition 2.1, and for $t \in [0, \infty)$, a convolution operator $\mathcal{K}_\ell(t) : \mathcal{U}\left(\Omega; \mathbb{R}^{d_v}\right) \to \mathcal{U}\left(\Omega; \mathbb{R}^{d_v}\right)$ with kernel $\kappa_\ell(t) : \mathbb{R}^n \to \mathbb{R}^{d_v \times d_v}$, a linear transform $W_\ell(t) : \mathbb{R}^{d_v} \to \mathbb{R}^{d_v}$, $R_\ell(t) = \mathcal{F}(\kappa_\ell(t)) : \mathbb{R}^n \to \mathbb{C}^{d_v \times d_v}$, $\varphi_\ell(t) : \mathbb{R}^n \to \mathbb{C}$, and $\psi_\ell(t) \in \mathbb{R}^{d_v \times d_v}$ the $\ell$-th continuous-time Fourier layer is defined as follows:*

$$\begin{aligned} \mathcal{L}_\ell(v)(t, x) &= \sigma\left(W_\ell(t)v(x) + \mathcal{K}_\ell(t)(v)(x)\right) \\ &= \sigma\left(W_\ell(t)v(x) + \mathcal{F}^{-1}\left(R_\ell(t) \cdot (\mathcal{F}v)\right)(x)\right) \\ &= \sigma\left(W_\ell\psi_\ell(t)v(x) + \mathcal{F}^{-1}\left(\varphi_\ell(t)R_\ell \cdot (\mathcal{F}v)\right)(x)\right), \ \forall x \in \Omega, \end{aligned} \tag{4}$$

**Time Modulating**   To equip the FNO network with the ability to capture information on the time of observations, a time-dependent layer (4) is constructed as follows. Two sharing networks, parameterized by two-layer fully connected networks together with sinusoidal embedding (Vaswani et al., 2017), first convert the input time $t$ into multi-dimensional representations $\varphi(t), \psi(t) \in \mathbb{R}^c$ for a hidden dimension $c$. For notational simplicity, we denote $\varphi_\ell(t, \xi) = \varphi_\ell(t)(\xi)$ and similar for $R(t, \xi)$. For each Fourier layer $\mathcal{L}_\ell$ and frequency $\xi$, learnable $A_\ell : \mathbb{R}^n \to \mathbb{C}^c$ and $B_\ell \in \mathbb{R}^{d_v \times c}$ produce time information $\varphi_\ell(t, \xi) = \varphi(t)^T A_\ell(\xi) \in \mathbb{C}$ and $\psi_\ell(t) = \text{diag}(B_\ell\psi(t)) \in \mathbb{R}^{d_v \times d_v}$ (the diagonal matrix with the elements of vector $B_\ell\psi(t)$ on the main diagonal). For each Fourier mode $\xi$, $R_\ell(\xi)$ is parameterized by a $(d_v \times d_v)$-complex matrix, and the time-dependent $R_\ell(t)$ is given by multiplying a complex number $\varphi_\ell(t, \xi)$, i.e., $R_\ell(t, \xi) = \varphi_\ell(t, \xi)R_\ell(\xi)$. In addition, $W_\ell(t)$ is parameterized by multiplication of two real $(d_v \times d_v)$-matrices $W_\ell$ and $\psi_\ell(t)$. This is somewhat reminiscent of weight modulation in StyleGAN2 (Karras et al., 2020), but applied per-frequency in the Fourier domain and per-channel in the spatial domain.

**Ablation study for time-aware network**   We provide insights into our model by investigating how useful the structure of the CTFNO designed based on the theoretical time-dependent Green's formula (3) is for learning time-dependent PDEs. We study three alternative ways to equip the temporal information into the FNO network, rather than **weight** modulation:

Baseline 1:   concatenating time $t$ with an **input** function $u_0(x)$.

Baseline 2:   concatenating encoded time $\varphi_0(t)$ with a **lifted input** function $v_0(x)$.

Baseline 3:   concatenating encoded times $\varphi_0(t), \varphi_1(t), \dots, \varphi_L(t)$ with intermediate **features** $v_0(x), v_1(x), \dots, v_L(x)$, respectively.

Table 1: Relative $L^2$ ($\times 10^{-2}$) errors of ablation studies.

| Model | Heat | Burgers |
|---|---|---|
| Baseline 1 | 11.60 | 19.63 |
| Baseline 2 | 2.84 | 18.68 |
| Baseline 3 | 2.68 | 18.92 |
| FNOseq | 75.61 | 73.10 |
| FNO2d | 0.58 | 10.32 |
| CTFNO | **0.22** | **5.68** |

See Appendix D.1 for more details. we test the ability of these models to learn canonical linear and nonlinear PDEs: one-dimensional heat and Burgers' equations. Table 1 show that, in all cases, the three ablation study models, which are structures that can continuously handle arbitrary time $t$, are similar to or worse than FNO2d (proposed in the original FNO paper) that can only evaluate the solution value at times on a specific grid. However, CTFNO significantly outperforms other models. The rationale of the way of imposing temporal information on CTFNO is based on Green's formula of time-dependent PDEs. The results validate that the time modulating structure of CTFNO designed based on the time-dependent Green's formula is much more expressive in actually learning the PDE.

### 3.3 UNIVERSAL APPROXIMATION

The universality of FNO has been proven by Kovachki et al. (2021). Based on this, we prove the universality of the proposed CTFNO, which is condensed in the following informal statement. A formal statement and details of proof of Theorem 3.1 are provided in Appendix A.

**Theorem 3.1. (Informal)** *CTFNO can approximate any time-dependent continuous operator, of arbitrary accuracy.*

### 3.4 STABILITY

When we learn a model for the time evolution of dynamical systems, the learned dynamics should be guaranteed to be well-posed. This is important because the learned system can be unstable when using a generic neural network (Szegedy et al., 2013; Moosavi-Dezfooli et al., 2017). In this case, the resultant output of the learned dynamics may not be continuous with respect to its input. Also, such unstable networks are vulnerable to adversarial attacks, overfitting, and unauthorized exploitation, which may render the network useless in practice. Therefore, stability is a necessary condition in real-world applications. The stability of a model is measured by the sensitivity of the prediction with respect to small perturbations of the inputs. A formal definition is given as follows.

**Definition 3.2.** *(Stability (Hadamard, 1902)) A PDE (2) is said to be stable if for any solution $u(t, x)$ with initial condition $u_0(x)$ and $\epsilon > 0$, there exists $\delta > 0$ such that for all new initial function $\tilde{u}_0(x)$ satisfying $\|\tilde{u}_0 - u_0\| < \delta$, the corresponding solution $\tilde{u}(t, x)$ satisfies $\|\tilde{u} - u\| < \epsilon$ for all $t \geq 0$.*

The stability is a hard constraint imposed upon the model. While some recent studies have begun to address the stability of neural network architectures, it has typically been used as a soft constraint by adding an extra regularization loss term (Moosavi-Dezfooli et al., 2019), or required computation of the eigenvalues of the Jacobian matrix (Ross & Doshi-Velez, 2018; Hoffman et al., 2019). Besides, stability of Neural ODEs are elicited by stable discretization techniques of the ODEs (Haber & Ruthotto, 2017; Yan et al., 2020). Similarly, the stability of CTFNO is connected to the well-posedness of the learned PDE, whose solution operator is the operator learned by CTFNO. The kernel formulation (3) sheds light on a way to ensure stability. A proof is deferred to appendix B.

**Proposition 3.2.** *(Stability of CTFNO) If $\|R(t, \xi)\|_2$ and $\|W(t)\|_2$ are bounded for every $t > 0$, then the corresponding CTFNO with a Lipschitz continuous activation function is stable.*

**Gershgorin discs normalization**   As given in proposition 3.2, the global stability of CTFNO is guaranteed if the Fourier kernel and weight have bounded $L^2$ norm. Because of its expensive computational cost, however, we suggest a practical method for enforcing stability conditions. Gershgorin's circle theorem (Varga, 2010) allows us to make fast deductions on the bound of eigenvalues. It states that every eigenvalue of a square matrix $A$ lies within the union of discs centered at diagonal elements with a radius of the sum of the magnitudes of the off-diagonal entries from the same row. Therefore, as we normalize the $L^1$ norm of each row, we can impose the requisite stability.

## 4 EXPERIMENTS

In this section, we empirically validate the applicability of the proposed model to study a variety of synthetic, and real applications. In all experiments, we build models with comparable network sizes to make a fair comparison. Precise implementation details can be found in Appendix C.

### 4.1 SYNTHETIC DATA

In this section, we use several illustrative examples to demonstrate the outstanding capacity of the proposed method in learning diverse class of dynamics. In all experiments, every model was trained with mean squared error (MSE) loss. We evaluate the performance of CTFNO in comparison with several related lines of work on a broad range of differential equations:

**Datasets**   We compare the performance of CTFNO on a broad range of synthetic data. Mathematical formulation of these dynamics can be found in Appendix C.1.

Table 2: RMSE ($\times 10^{-2}$) results on synthetic data.

| Model | Stiff | Spiral | Sawtooth | Square | Reaction | Heat | Burgers |
|---|---|---|---|---|---|---|---|
| NODE | 44.23 | 3.26 | 28.09 | 97.50 | 5.109 | 0.351 | 5.348 |
| ANODE | 37.18 | 3.26 | 9.92 | 80.95 | 4.241 | 0.351 | 4.214 |
| Neural Flow | 26.39 | 3.26 | 7.93 | 20.40 | 10.300 | 3.038 | 12.602 |
| Neural Laplace | 20.83 | 4.25 | 4.84 | 17.06 | 2.804 | 2.205 | 5.291 |
| DON | - | - | - | - | 3.055 | 0.473 | 6.022 |
| PDN | - | - | - | - | 3.316 | 0.3234 | 5.796 |
| FNO2d | - | - | - | - | 0.472 | 0.0330 | 3.136 |
| CTFNO | **16.53** | **1.87** | **3.90** | **11.69** | **0.239** | **0.0269** | **1.952** |

- **Stiff ODE** (Holt et al., 2022) is a typical example that Neural ODEs fail to learn.

- **Spiral ODE** is a two-dimensional system of nonlinear ODEs, commonly arising in biological systems. The dynamics describe spiral shaped trajectories.

- **Sawtooth and Square** (Biloš et al., 2021) are considered to evaluate the capability on modeling waveform signals. These are piecewise differentiable having cusps.

- **Reaction ODE** is commonly used to model chemical reactions and is of the form $\partial u / \partial t = 6u(1 - u)$. Unlike to aforementioned ODEs, a solution to Reaction ODE is regarded as a function and in implementation it can be regarded as a high-dimensional ODE.

- **Heat** (Baron Fourier, 1878) **and Burgers' equations** (Bateman, 1915) are canonical one-dimensional linear and nonlinear PDEs. They take the form

$$\frac{\partial u}{\partial t} + \alpha u \frac{\partial u}{\partial x} = \nu \frac{\partial^2 u}{\partial x^2}, \quad x \in (0, 1), \ t \in (0, T], \tag{5}$$

with the corner cases: heat $\alpha = 0$ and Burgers' equation $\alpha = 1$. Given an initial function $u_0$, we train models to learn $(t, u_0(x)) \mapsto u(t, x)$ for $t \in (0, T]$.

**Baselines** We evaluate the performance of our model in comparison with several related lines of works: the standard Neural ODE (NODE) (Chen et al., 2018), ANODE (Dupont et al., 2019), and Neural Flow (Biloš et al., 2021), which directly parametrizes the solution operator of an ODE, are adopted for ODE-based models. We also consider with Neural Laplace (Holt et al., 2022) which can represent diverse classes of equations by modeling them in the Laplace domain. Furthermore, we compare with representative PDE-based models; DeepONet (DON) (Lu et al., 2019a), an alternative operator learning method, POD-DeepONet (PDN) (Lu et al., 2022), a functional PCA-based model, and FNO2d (Li et al., 2021), a Fourier neural operator with spatio-temporal inputs.

**Results Discussion** The overall results of root mean squared error (RMSE) are reported in Table 2. The results show that our CTFNO substantially achieves conspicuous success all baseline models by a large margin on all datasets. We can see that the performance difference of baselines on different datasets varies a lot. An ODE model that performs well on a certain data fails to learn the dynamics of another data. But, Neural Laplace, which can express wider dynamics than ODE-based models, performs better than the three ODE-based models (NODE, ANODE, and Neural Flow) throughout all datasets. Further, our model, which can describe a broad family of DEs, consistently approximate all synthetic dynamics much better. These results demonstrate how important the range of expressible dynamics of the model is. Figure 1 provides pictorial results of the learned solutions to Spiral, Square, and Sawtooth data, and further quatlitative comparisons are included in Appendix D.5. Table 2 also contain the comparison to the original FNO2d. Comparison results of CTFNO with the original FNO demonstrate that the use of the proposed time-dependent structure significantly improves the capacity of the model to handle time. Moreover, our model is superior to existing benchmark PDE models. The overall results confirm that CTFNO is amenable to learn a wide array of dynamics.

## 4.2 REAL TIME-SERIES APPLICATIONS

In this section, we investigate the performance of our method on interpolation and prediction tasks using three real-world time-series datasets, including partially observed, multi-variate sequences. All models were run three times with different random seeds and we report the averaged value. Results show that the CTFNO performs outperforms a range of baselines.

Table 3: MSE ($\times 10^{-3}$) on the MuJoCo dataset.

| Model | Interpolation (% Observed Points) | | | | Prediction (% Observed Points) | | | |
|---|---|---|---|---|---|---|---|---|
| | 10% | 20% | 30% | 50% | 10% | 20% | 30% | 50% |
| RNN-VAE | 65.14 | 64.08 | 63.05 | 61.00 | 23.78 | 21.35 | 20.21 | 17.82 |
| ODE-RNN | 16.47 | 12.09 | 9.86 | 6.65 | 135.08 | 319.5 | 154.65 | 264.63 |
| LODE (RNN enc) | 24.77 | 5.78 | 27.68 | 4.47 | 16.63 | 16.53 | 14.85 | 13.77 |
| LODE (ODE enc) | 3.60 | 2.95 | 3.00 | 2.85 | 14.41 | 14.00 | 11.75 | 12.58 |
| Neural Flow | 7.15 | 5.58 | 4.96 | 4.60 | 17.99 | 16.10 | 15.48 | 15.29 |
| CTFNO | **1.53** | **1.18** | **1.12** | **1.15** | **9.26** | **8.93** | **8.42** | **8.70** |

**Datasets**  We evaluate our model on MuJoCo hopper, PhyisioNet 2012, and Human Activity datasets, as in Rubanova et al. (2019). For more details, see Appendix C.1.

- **MuJoCo** (Tassa et al., 2018) We consider the hopper physics environments from the Deepmind Control Suite. Data records 14-dimensional attributes, including state and action, with 100 timestamps. To deal with partial observations, we conduct interpolation and prediction tasks, in which we reveal either 10%, 20%, 30%, or 50% of the ground truth.

- **PhysioNet 2012** (Silva et al., 2012) To investigate sparsely observed time series, we evaluate our model on a real-world clinical dataset. PhysioNet Challenge 2012 dataset is irregularly sampled and sparsely observed. The goal is to interpolate and predict 41 biomedical features, such as heart rate and glucose, of intensive care unit (ICU) patients.

- **Human Activity** Human Activity dataset consists of sporadically observed sensor data collected from five individuals performing several activities (i.e. walking, standing, laying, etc). We use the Activity dataset together with all pre-processing steps as they were provided by (Rubanova et al., 2019), resulting in 6554 sequences of 211 time points. We train our model to classify the type of human activities from sequential data.

**Baselines**  Five models are chosen for the comparisons: (1) RNN-VAE, a variational autoencoder (VAE) (Kingma & Welling, 2014; Rezende et al., 2014) model whose encoder and decoder are recurrent neural networks (RNNs). (2) ODE-RNN (Rubanova et al., 2019), a RNN model which uses Neural ODEs to model hidden state dynamics. (3) Two Latent ODE (LODE) models with RNN (Chen et al., 2018) and ODE-RNN (Rubanova et al., 2019) encoders. (4) Coupling Flow in Neural Flow (Biloš et al., 2021), a neural operator which directly models the solution curves of an ODE.

**Results Discussion**  We follow the encoder-decoder framework of Latent ODEs that leverages a VAE architecture to represent incomplete time-series data as a continuous-time model. For CTFNO, we employ a simple RNN encoder, and focus on the model representations of inherent dynamics in the latent space. Results on MuJoCo reported in Table 3 show that CTFNO consistently outperforms the baseline models to a large extent on all tasks of both interpolation and prediction across all kinds of observed time points. This validates that our model can be used as a relevant model for real applications on time-series with missing time steps. Table 4 summarizes interpolative and predictive performance on PhysioNet, and per-time-point classification accuracy on Activity. CTFNO is superior to all of the benchmark models on both PhysioNet and Activity, which indicate that our model provides a useful utilization of sparsely observed time-series data and a meaningful representation for classification. The overall results demonstrate that our proposed model, which is based on PDEs, consistently improves the performance of ODE-based models in real applications and is a novel approach that can handle a wide range of real-world problems.

Table 4: MSE ($\times 10^{-3}$) on PhysioNet and per-time-point classification accuracies (%) on Activity.

| Model | PhysioNet | | Activity |
|---|---|---|---|
| | Interpolation | Prediction | Accuracy |
| RNN-VAE | 5.93 | 3.05 | 34.3 |
| ODE-RNN | 2.36 | - | 82.9 |
| LODE (RNN enc) | 3.16 | 5.78 | 83.5 |
| LODE (ODE enc) | 2.23 | 2.95 | 84.6 |
| Neural Flow | 2.94 | - | 65.7 |
| CTFNO | **1.80** | **2.10** | **85.0** |

Figure 3: Effect of stabilization on the Plane Vibration dataset.

## 4.3 STUDIES ON STABILITY

This section is devoted to validating the effect of the proposed stabilization scheme in Section 3.4. We include experimental results on a real-time series data and adversarial attacks on image classification.

**Datasets**   We the stability on a real-world time-series data and two image datasets.

- **Plane Vibration** (Noël & Schoukens, 2017) is considered to elucidate how stabilization scheme works in practical time-series data. It is a multi-variate data, consisting of five features; force, voltage, and accelerations measured in three spots. We comply the experimental setup as provided in HBNODEs (Xia et al., 2021). The task of plane vibration is interested in forecasting the next eight time steps from the previous 64 consecutive time observations.

- **Image datasets** We perform experiments on image classification, which is one of the benchmark applications of Neural ODEs, on two benchmark datasets: MNIST (LeCun et al., 1998) and CIFAR10 (Krizhevsky et al., 2009). Although this is not a time-dependent problem, we include classification to validate the effect of the proposed stabilization scheme. Images could be considered as functions $a : [0, 1]^2 \to \mathbb{R}^3$ of light defined on a continuous region of pixel locations, instead of $32 \times 32$ pixel vectors of RGB values. Classification requires us to learn a dynamics that simultaneously drives each input $a(\mathbf{x})$ to the corresponding final feature $u(\mathbf{x})$, allocated depending on its label. Therefore, posing the problem as learning an operator that maps the image $a(\mathbf{x})$ to $u(\mathbf{x})$, we can apply FNO to image classification.

**Baselines**   We compare the performance on plane vibration with the results reported in (Xia et al., 2021), which contains second-order Neural ODE models including ANODE, SONODE (Norcliffe et al., 2020), HBNODE, and GHBNODE (Xia et al., 2021). For image classification, we take CNN and ANODE as baselines. All models are fit to share the similar number of parameters.

**Improvement in Model Generalization**   MSE ($\times 10^{-2}$) of Neural ODEs reported in (Xia et al., 2021) is $\geq 2.5$, and it is annotated by a green dashed line in Figure 3. Our CTFNO achieves much lower MSE of 1.96. Figure 3 presents a positive effect of stabilization; a tendency of CTFNO to be less prone to overfitting. The ability to generalize well outside the training dataset is essential for models to be practically useful. We observe that increasing the Gershgorin's discs stabilization parameter $M$ leads to overfitting, and too small $M$ produces a degenerated system, dropping the performance. Besides, the model with congenial stabilization does not suffer from overfitting or degradation of the accuracy, which validates the effect of stabilization. Moreover, it is notable that Neural ODEs require tremendous computational burdens due to their use of numerical ODE solvers. By obviating the need for the ODE solver, the computational time of CTFNO is merely about 10% of Neural ODEs. Similar investigation in image classification is included in Appendix D.2.

Table 5: Classification accuracy (%) and robustness of different models on MNIST.

| Model | Clean | Gaussian noise | | Adversarial attack | | | |
|---|---|---|---|---|---|---|---|
| | | $\sigma = 50$ | $\sigma = 100$ | FGSM-30/255 | FGSM-50/255 | PGD-30/255 | PGD-50/255 |
| CNN | **99.0** | 29.7 | 10.4 | 27.3 | 15.3 | 10.8 | 4.6 |
| ANODE | 97.9 | 91.8 | 56.4 | 26.9 | 1.5 | 62.5 | 9.7 |
| FNO | 98.6 | 90.2 | 45.6 | 36.9 | 25.8 | 8.1 | 4.1 |
| FNO (WD) | 98.9 | 90.0 | 43.6 | **57.6** | **35.6** | 22.1 | 11.7 |
| FNO (Stab) | 98.8 | **97.8** | **88.2** | 36.2 | 3.43 | **62.6** | **47.1** |

Table 6: Classification accuracy (%) and robustness of different models on CIFAR10.

| Model | Clean | Gaussian noise | | Adversarial attack | | | |
|---|---|---|---|---|---|---|---|
| | | $\sigma = 15$ | $\sigma = 20$ | FGSM-2/255 | FGSM-5/255 | PGD-2/255 | PGD-5/255 |
| CNN | **75.5** | 60.0 | 51.1 | 23.0 | 4.7 | 16.7 | 0.43 |
| ANODE | 59.4 | 40.9 | 34.9 | 0.1 | 0.0 | 0.0 | 0.0 |
| FNO | 64.0 | 57.9 | 52.2 | 3.4 | 1.4 | 1.5 | 0.0 |
| FNO (WD) | 66.9 | 54.9 | 46.2 | 5.8 | 0.6 | 2.0 | 0.0 |
| FNO (Stab) | 65.5 | **64.3** | **63.3** | **38.6** | **11.1** | **38.1** | **9.1** |

**Robustness on Adversarial Attack**   By ensuring the stability of the learned PDE in Section 3.4, we may expect that the resulting solution for an input with a small perturbation converges to the same label as the unperturbed one. To investigate this stabilization effect on FNO, we test the trained model in defending against Gaussian noise, fast gradient signed method (FGSM) (Goodfellow et al., 2014), and projected gradient descent (PGD) (Madry et al., 2017) attack. Weight decay (WD) is a standard practical technique to regularize the $L^2$ norm of weights of models. Because WD has a similar intention to our stabilization scheme, these two are also compared. For fair comparison, CNN and ANODE are trained with WD. Tables 5 and 6 show that for both datasets, our stabilization significantly enhances the robustness of the model. Our model achieves better results in both accuracy and robustness than ANODE. Although CNN produces around 10% higher accuracy in CIFAR10, our model outperforms CNN for perturbed images. Moreover, the results indicate that FNO trained with WD is prone to mislead by adversarial attacks. This demonstrates that it is much more accurate to apply a model-suitable stabilization based on the PDE theory, rather than naively regularizing the $L^2$ norm of the weights.

## 5 RELATED WORK

**ODE based Networks**   The interpretation of a residual network (He et al., 2016) as a discretization of an ODE (Chen et al., 2018; Haber & Ruthotto, 2017; Lu et al., 2018) provided an interface between deep learning and ODEs. Subsequently, extensive work has been conducted on parametrizing the continuous dynamics of hidden states using an ODE specified by neural networks (Greydanus et al., 2019; Lu et al., 2019b; Liu et al., 2021). These approaches have found wide applicability across diverse applications including vision tasks (Scao, 2020; Park & Kim, 2021; Pinckaers & Litjens, 2019; Xu et al., 2021), mean-field games (Ruthotto et al., 2020), speech synthesis (Kim et al., 2020a), and density estimation (Chen et al., 2019; Yildiz et al., 2019). Owing to the continuous representation of neural networks, Neural ODEs are particularly attractive for irregularly-sampled time series data (Rubanova et al., 2019; De Brouwer et al., 2019; Chang et al., 2019; Kidger et al., 2020; Chen et al., 2020). A recent work(Biloš et al., 2021) circumvents the usage of an expensive numerical integration by directly parametrize the solution trajectory of an ODE. Furthermore, stable (Kolter & Manek, 2019; Zhang et al., 2019; Kang et al., 2021; Tuor et al., 2020) and reversible (Chang et al., 2018; Grathwohl et al., 2018) architectures has been developed based on the intrinsic properties of ODEs.

**PDE based Networks**   Some studies have integrated PDEs and neural networks (Eliasof et al., 2021; Ruthotto & Haber, 2020; Ben-Yair et al., 2021; Sun et al., 2020). These works have established based on a link between convolution filters in CNNs and differential operators (Long et al., 2018), or on a finite difference discretization (Strikwerda, 2004) of derivatives. Recently, a new approach (Kim et al., 2020b) that uses a physics-informed neural network (PINN) (Raissi et al., 2019) has been proposed. All of these works assign author-defined specific PDEs to neural networks. Because the nature of each problem to which the model be applied differ considerably, the choice of dynamics has been shown to be quite important. Therefore, such architectures designed with fixed PDEs cannot be widely applied. Despite their PDE-based motivation, those approaches were only applied to image classification tasks and did not explore the application on continuous inputs and time-series data.

## 6 CONCLUSION & LIMITATIONS

In this paper, we presented a novel approach for modeling time-series in terms of PDEs. As time is intrinsically continuous, we proposed a neural operator CTFNO that learns the underlying PDE by equipping FNO with the ability to represent time in a continuous manner. We also provided theoretical guarantees for the universal approximation and the stability of CTFNO. Our comprehensive experiments demonstrated that CTFNO outperforms existing differential equation based models on synthetic and real-world datasets, and the proposed stabilization method effectively improves robustness and prevents overfitting.

We note that FNO is hard to embody discontinuous features because Fourier transform only captures global information. We expect that extending our approach to a model that can extract local features, such as (Gupta et al., 2021), provides interesting avenues for future work. Also, the stabilization parameter $M$ is a user-defined hyperparameter. The optimal $M$ for a given data is unknown and this is one of our limitation.

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

# A  UNIVERSAL APPROXIMATION

## A.1  MATHEMATICAL NOTATION

We introduce the symbols and mathematical notations that are frequently used in this paper.

| Symbol | Description |
| --- | --- |
| $\sigma$ | activation function |
| $d$ | spatial dimension of domain |
| $\mathbb{T}^d$ | periodic torus, identified with $[0, 2\pi]^d$ |
| $d_a$ | dimension of the input $a(x)$ |
| $d_u$ | dimension of the output solution $u(x)$ |
| $d_v$ | dimension of the augmented representation $v(x)$ |
| $T$ | terminal time |
| $L$ | the number of layers |
| $\Omega \subset \mathbb{R}^d$ | spatial domain |
| $x$ | point in the the spatial domain |
| $\xi$ | frequency variable in the Fourier domain |
| $\mathcal{A}(D; \mathbb{R}^{d_a})$ | input function space |
| $\mathcal{U}(D; \mathbb{R}^{d_u})$ | output function space |
| $\mathcal{F}_X, \mathcal{F}_X^{-1}$ | Fourier transform and inverse Fourier transform with respect to $X$ |
| $\mathcal{F}_N, \mathcal{F}_N^{-1}$ | discrete Fourier transform and inverse |
| $\mathcal{K}_N$ | Fourier wavenumbers $\mathcal{K}_N = \left\{ \xi \in \mathbb{Z}^d \,\middle|\, |\xi|_\infty \le N \right\}$ |
| $\mathcal{P}$ | lifting operator |
| $\mathcal{L}_\ell$ | neural operator layer |
| $\mathcal{Q}$ | projection operator |
| $L^2$ | space of square-integrable functions |
| $L_N^2$ | $L_N^2 \subset L^2$ trigonometric polynomials of degree $\le N$ |
| $H^s$ | Sobolev space of smoothness $s$, with norm $\|\cdot\|_{H^s}$ |
| $P_N$ | $L^2$-orthogonal Fourier projection $P_N : L^2 \to L_N^2$ |
| $C([0,T] \times X)$ | space of continuous functions $u : [0,T] \to X$ satisfying $\|u\|_{C([0,T] \times X)} = \max\limits_{0 \le t \le T} \|u(t)\| < \infty$ |
| $\mathbf{D}_n$ | space of $n \times n$ diagonal matrices |
| $I_n$ | $n \times n$ identity matrix |
| $\odot$ | component-wise matrix multiplication |

## A.2  PROOF OF UNIVERSAL APPROXIMATION

A neural operator

$$\mathcal{N} : \mathcal{A}\left(\Omega; \mathbb{R}^{d_a}\right) \to C\left([0,T] \times \mathcal{U}\left(\Omega; \mathbb{R}^{d_u}\right)\right)$$
$$a \mapsto \mathcal{N}(a) \tag{6}$$

is defined as the following form

$$\mathcal{N}(a) = \mathcal{Q} \circ \mathcal{L}_L \circ \cdots \mathcal{L}_1 \circ \mathcal{P}(a) \tag{7}$$

with a lifting operator

$$\mathcal{P} : \mathcal{A}\left(\Omega; \mathbb{R}^{d_a}\right) \to C\left([0,T] \times \mathcal{U}\left(\Omega; \mathbb{R}^{d_v}\right)\right)$$
$$\mathcal{P}(a)(t,x) := Pa(x), \qquad\qquad P \in \mathbb{R}^{d_v \times d_a}$$

for all $a \in \mathcal{A}\left(\Omega; \mathbb{R}^{d_a}\right)$, $x \in \Omega$, and a projection operator

$$\mathcal{Q} : C\left([0,T] \times \mathcal{U}\left(\Omega; \mathbb{R}^{d_v}\right)\right) \to C\left([0,T] \times \mathcal{U}\left(\Omega; \mathbb{R}^{d_u}\right)\right)$$
$$\mathcal{Q}(v)(t,x) := Qv(t,x), \qquad\qquad Q \in \mathbb{R}^{d_u \times d_v}.$$

Moreover, as introduced in Section 3.2 the time-dependent neural operator layer is defined by

$$
\begin{aligned}
&\mathcal{L}_\ell(v)(t,x) \\
&:= \sigma\left(W_\ell(t)v(t,x) + b_\ell(t,x) + (\mathcal{K}(a;\theta_\ell)v)(t,x)\right) \\
&:= \sigma\left(W_\ell(t)v(t,x) + b_\ell(t,x) + \mathcal{F}_X^{-1}\left(R_\ell(t,\xi)\cdot\mathcal{F}_X(v)(\xi)\right)(x)\right) \\
&:= \sigma\left(W_\ell\psi_{w,\ell}(t)v(t,x) + \psi_{b,\ell}(t)b_\ell(x) + \mathcal{F}_X^{-1}\left(\varphi_\ell(t,\xi)R_\ell(\xi)\cdot\mathcal{F}_X(t,v)(\xi)\right)(t,x)\right).
\end{aligned}
\tag{8}
$$

Before we establish a proof of the universality of CTFNO, let us start by recalling the following Lemma provided in (Kovachki et al., 2021).

**Lemma A.1** ((Kovachki et al., 2021)). *For FNO introduced in Section 2, the following holds.*

(a) *Let $\mathcal{G}: H^s\left(\mathbb{T}^d;\mathbb{R}^{d_a}\right) \to H^{s'}\left(\mathbb{T}^d;\mathbb{R}^{d_u}\right)$ be continuous operator with $s, s' \geq 0$ and $K \subset H^s$ be a compact subset. Then for any $\epsilon > 0$ there exists $N \in \mathbb{N}$, such that for all $a \in K$*

$$
\|\mathcal{G}(a) - \mathcal{G}_N(a)\|_{L^2} < \epsilon,
$$

*where $\mathcal{G}_N: H^s\left(\mathbb{T}^d\right) \to L^2\left(\mathbb{T}^d\right)$ is defined by $\mathcal{G}_N(a) := P_N\mathcal{G}(P_Na)$.*

(b) *Let $B > 0$ and $\epsilon > 0$ be given. Then there exists a FNO $\mathcal{N}_{FT}$ such that for all $v \in L^2$ with $\|v\|_{L^2} \leq B$*

$$
\|\mathcal{F}_N \circ P_N v - \mathcal{N}_{FT}v\|_\infty < \epsilon.
$$

(c) *Let $B > 0$ and $\epsilon > 0$ be given. Then there eixsts a FNO $\mathcal{N}_{IFT}$ such that for all $\{\hat{v}(\xi) : \xi \in \mathcal{K}_N\}$ with $\|\hat{v}\|_{L^\infty(\mathcal{K}_N)} \leq B$*

$$
\left\|\mathcal{F}_N^{-1}\left(\{\hat{v}(\xi)\}\right) - \mathcal{N}_{IFT}\left(\{\hat{v}(\xi)\}\right)\right\|_{L^2} < \epsilon.
$$

(d) *Let $B > 0$ and $\epsilon > 0$ be given. Then there exists a FNO $\widehat{\mathcal{N}}$ such that for any $\{\hat{v}(\xi)\} \in \mathbb{C}^{\mathcal{K}_N}$ with $\|\hat{v}\|_{L^\infty(\mathcal{K}_N)} \leq B$ such that*

$$
\left\|\widehat{\mathcal{G}}_N\left(\{\hat{v}(\xi)\}\right) - \widehat{\mathcal{N}}\left(\{\hat{v}(\xi)\}\right)\right\|_\infty < \epsilon.
$$

In what follows, we prove our main theorem showing that the proposed CTFNO is a universal approximator of continuous operators.

**Theorem A.2** (Universal approximation of CTFNO). *Let $s, s' \geq 0$. Suppose $\mathcal{G}: H^s\left(\mathbb{T}^d;\mathbb{R}^{d_a}\right) \to C\left([0,T] \times H^{s'}\left(\mathbb{T}^d;\mathbb{R}^{d_u}\right)\right)$ is a continuous operator and $K \subset H^s\left(\mathbb{T}^d;\mathbb{R}^{d_a}\right)$ is a compact subset. Then for any $\epsilon > 0$, there exists a CTFNO $\mathcal{N}: H^s\left(\mathbb{T}^d;\mathbb{R}^{d_a}\right) \to C\left([0,T] \times H^{s'}\left(\mathbb{T}^d;\mathbb{R}^{d_u}\right)\right)$, of the form (7), continuous as an operator $H^s\left(\mathbb{T}^d;\mathbb{R}^{d_a}\right) \to C\left([0,T] \times H^{s'}\left(\mathbb{T}^d;\mathbb{R}^{d_u}\right)\right)$, such that*

$$
\sup_{a \in K} \|\mathcal{G}(a) - \mathcal{N}(a)\|_{C\left([0,T] \times H^{s'}\left(\mathbb{T}^d;\mathbb{R}^{d_u}\right)\right)} < \epsilon.
$$

The proof is essentially split into two parts. The first part states that the Fourier projection can approximate continuous operators. To simplify notation, we use shorthand $H^s = H^s\left(\mathbb{T}^d;\mathbb{R}^{d_a}\right)$, $H^{s'} = H^{s'}\left(\mathbb{T}^d;\mathbb{R}^{d_u}\right)$ and $a_t(x) := a(t,x)$ in the remainder of the paper.

**Lemma A.3.** *Let $\mathcal{G}: C\left([0,T] \times H^s\right) \to C\left([0,T] \times H^{s'}\right)$ be a continuous operator and $K \subset C\left([0,T] \times H^s\right)$ a compact subset. Then for any $\epsilon > 0$ there exists $N \in \mathbb{N}$, such that for all $a \in K$ and $t \in [0,T]$*

$$
\|\mathcal{G}(a)_t - \mathcal{G}_N(a)_t\|_{H^{s'}} < \epsilon.
$$

*Proof.* Since $\mathcal{G}$ is uniformly continuous on a compact domain $\mathcal{G}(K)$, for any $\epsilon > 0$ there exists $\delta > 0$ such that

$$
\|\mathcal{G}(a)_{t_1} - \mathcal{G}(a)_{t_2}\|_{H^{s'}} < \epsilon, \quad \forall a \in K,
$$

provided $|t_1 - t_2| < \delta$. Let $0 = t_0 < t_1 < \ldots < t_M = T$ be a partition of $[0, T]$ with $t_{i+1} - t_i < \delta$ and $\mathcal{G}_i : H^s \to H^{s'}$ be defined by $a \mapsto \mathcal{G}(a)_{t_i}$ for each $i$. Then, (a) in Lemma A.1 furnishes that there exists $N_i \in \mathbb{N}$ such that

$$\|\mathcal{G}(a)_{t_i} - \mathcal{G}_{N_i}(a)_{t_i}\|_{H^{s'}} < \epsilon, \quad \forall a \in K.$$

Combining all, for $N > \max\{N_i : i = 0, \ldots, M\}$, we have

$$\begin{aligned}
&\|\mathcal{G}(a)_t - \mathcal{G}_N(a)_t\|_{H^{s'}} \\
&\leq \|\mathcal{G}(a)_t - \mathcal{G}(a)_{t_i}\|_{H^{s'}} + \|\mathcal{G}(a)_{t_i} - \mathcal{G}_N(a)_{t_i}\|_{H^{s'}} + \|\mathcal{G}_N(a)_{t_i} - \mathcal{G}_N(a)_t\|_{H^{s'}} \\
&< 3\epsilon,
\end{aligned}$$

for any $a \in K$ and $t \in [0, T]$. This proves the lemma. $\qquad\square$

The second part is to prove that any time-dependent continuous function can be approximated by a three-layered feed forward network combined with the proposed time-module.

**Lemma A.4.** *Let $f : [0, T] \times \mathbb{R}^{d_x} \to \mathbb{R}^{d_y}$ be a continuous function and $K \subseteq \mathbb{R}^{d_x}$ a compact subset. Then for any $\epsilon > 0$, there exists a three-layered feed forward network with time*

$$\begin{aligned}
\mathcal{N} &:= \mathcal{Q} \circ \mathcal{L}_2 \circ \mathcal{L}_1, \\
\mathcal{L}_\ell(t, x) &:= (t, \sigma(\psi_{w,\ell}(t) W_\ell x + \psi_{b,\ell}(t) b_\ell)), \\
\mathcal{Q}(t, x) &= Qx,
\end{aligned}$$

*satisfying*

$$\|\mathcal{N}(t, x) - f(t, x)\| < \epsilon, \quad \forall (t, x) \in [0, T] \times K,$$

*where $\sigma$ is an activation function, $\psi_{w,\ell} : [0, T] \to \mathbf{D}_{d_v}$ and $\psi_{b,\ell} : [0, T] \to \mathbb{R}^{d_v}$ are continuous, $Q$ is a projection matrix, and $W_\ell, b_\ell$ are weight matrix and bias vector for each $\ell$, respectively.*

*Proof.* Without loss of generality, we may assume $d_y = 1$. Construct $W_1 \in \mathbb{R}^{(d_x+1) \times d_x}$ and $b_1 \in \mathbb{R}^{d_x+1}$ as

$$W_1 = \begin{pmatrix} \mathbf{I}_{d_x} \\ 0 \end{pmatrix}, b_1 = \begin{pmatrix} \mathbf{0}_{d_x} \\ 1 \end{pmatrix},$$

with $\psi_{w,1}(t) = \mathbf{0}_{d_x+1}$, $\psi_{b,1}(t) = t\mathbf{1}_{d_x+1}$. Then, $\mathcal{L}_1(t, x) = (t, (x, t)^T) \in [0, T] \times \mathbb{R}^{d_x+1}$. By the universal approximation theorem of Cybenko (1989), for any $\epsilon > 0$ there exists $W_2 \in \mathbb{R}^{d_v \times (d_x+1)}$, $b_2 \in \mathbb{R}^{d_v}$, and $Q \in \mathbb{R}^{1 \times d_v}$ such that

$$\left| Q\sigma\left(W_2 \begin{pmatrix} x \\ t \end{pmatrix} + b_2\right) - f(t, x) \right| < \epsilon, \quad \forall (t, x) \in [0, T] \times K.$$

By setting $\psi_{w,2} = \mathbf{I}_{d_v}$, $\psi_{b,2} = \mathbf{1}_{d_v}$, we have the desired $\mathcal{N}$. $\qquad\square$

With these lemmas out of the way, we are ready to provide a proof of Theorem 3.1.

For given continuous operator $\mathcal{G} : C([0, T] \times H^s) \to C([0, T] \times L^2)$, compact subset $K \subset C([0, T] \times H^s)$ and small $\epsilon > 0$, Lemma A.3 gives an integer $N \in \mathbb{N}$ such that

$$\sup_{a \in K} \|\mathcal{G}(a)_t - \mathcal{G}_N(a)_t\|_{L^2} < \epsilon,$$

for each time $t \in [0, T]$. Since $K$ is compact, there exists $B > 0$ satisfying $\|v_t\|_{L^2} \leq B$ for all $v \in K$ and $t \in [0, T]$. For given small $\delta \in (0, 1)$, let $\mathcal{N}_{FT}$ be a CTFNO such that for all function $w$ with $\|w\|_{L^2} \leq B$

$$\|\mathcal{F}_N \circ P_N w - \mathcal{N}_{FT} w\|_\infty < \delta.$$

In fact, $\delta$ is chosen according to the Lipschitz constant of $\widehat{\mathcal{N}}$, which will be constructed. As claimed by $(c)$ in Lemma A.1, there exists a CTFNO $\mathcal{N}_{IFT}$ such that for all $\{\hat{w}(\xi) : \xi \in \mathcal{K}_N\}$ with $|\hat{w}(\xi)| \leq \sup_{t \in [0,T]} \sup_{a \in K} \|\mathcal{F}_N \circ P_N(a_t)\|_\infty + 1 \leq B' = C(N) \sup_{t \in [0,T]} \sup_{a \in K} \|a_t\|_2 + 1$

$$\left\|\mathcal{F}_N^{-1}(\{\hat{w}(\xi)\}) - \mathcal{N}_{IFT}(\{\hat{w}(\xi)\})\right\|_{L^\infty} < \epsilon.$$

The continuity of $\mathcal{F}_N^{-1}$ and $\mathcal{N}_{IFT}$ leads to the existence of $B'' > 0$ that satisfies

$$\left\| \mathcal{F}_N^{-1} \left( \{\hat{w}(\xi)\} \right) \right\|_\infty \leq B'', \quad \left\| \mathcal{N}_{IFT} \left( \{\hat{w}(\xi)\} \right) \right\|_\infty \leq B'',$$

for all $\{\hat{w}(\xi) : \xi \in \mathcal{K}_N\}$ with $|\hat{w}(\xi)| \leq B''$. By (d) in Lemma A.1 and Lemma A.4, we can construct a CTFNO $\widehat{\mathcal{N}}$ such that for any $\{\hat{w}(\xi)\} \in \mathbb{C}^{\mathcal{K}_N}$ with $\|\hat{w}\|_{L^\infty} \leq B''$,

$$\sup_{t \in [0,T]} \left\| \widehat{\mathcal{G}}_N \left( t, \{\hat{w}(\xi)\} \right) - \widehat{\mathcal{N}} \left( t, \{\hat{w}(\xi)\} \right) \right\|_\infty < \epsilon.$$

Note that the construction of $\widehat{\mathcal{N}}$ is independent on $\delta$ and $\mathcal{N}_{FT}$. Now we set $\delta > 0$ so small that

$$\mathrm{Lip}\left( \widehat{\mathcal{N}} \right) \sup_{t \in [0,T]} \left\| \mathcal{F}_N \circ P_N(a_t) - \mathcal{N}_{FT}(a_t) \right\|_\infty < \epsilon,$$

for all $a \in K$, which implies that

$$\sup_{a \in K} \left\| \widehat{\mathcal{N}} \circ \mathcal{F}_N \circ P_N(a_t) - \widehat{\mathcal{N}} \circ \mathcal{N}_{FT}(a_t) \right\|_{L^2} < \epsilon.$$

Finally, for each $t \in [0,T]$ we can deduce the following inequality:

$$\begin{aligned}
\sup_{a \in K} & \left\| \mathcal{G}(a)_t - \mathcal{N}(a)_t \right\|_{L^2} \\
& < \epsilon + \sup_{a \in K} \left\| \mathcal{G}_N(a)_t - \mathcal{N}(a)_t \right\|_{L^2} \\
& = \epsilon + \sup_{a \in K} \left\| \mathcal{F}_N^{-1} \circ \widehat{\mathcal{G}}_N \circ \mathcal{F}_N \circ P_N(a_t) - \mathcal{N}_{IFT} \circ \widehat{\mathcal{N}} \circ \mathcal{N}_{FT}(a_t) \right\|_{L^2} \\
& \leq \epsilon + \sup_{a \in K} \left\| \mathcal{F}_N^{-1} \circ \widehat{\mathcal{G}}_N \circ \mathcal{F}_N \circ P_N(a_t) - \mathcal{F}_N^{-1} \circ \widehat{\mathcal{N}} \circ \mathcal{N}_{FT}(a_t) \right\|_{L^2} \\
& \quad + \sup_{a \in K} \left\| \mathcal{F}_N^{-1} \circ \widehat{\mathcal{N}} \circ \mathcal{N}_{FT}(a_t) - \mathcal{N}_{IFT} \circ \widehat{\mathcal{N}} \circ \mathcal{N}_{FT}(a_t) \right\|_{L^2} \\
& \leq \epsilon + \sup_{a \in K} \left\| \widehat{\mathcal{G}}_N \circ \mathcal{F}_N \circ P_N(a_t) - \widehat{\mathcal{N}} \circ \mathcal{N}_{FT}(a_t) \right\|_\infty \\
& \quad + \sup_{\substack{\{\hat{w}(\xi)\} \in \mathbb{C}^{\mathcal{K}_N} \\ \|\hat{w}(\xi)\|_\infty \leq B''}} \left\| \mathcal{F}_N^{-1} \left( \{\hat{w}(\xi)\} \right) - \mathcal{N}_{IFT} \left( \{\hat{w}(\xi)\} \right) \right\|_{L^2} \\
& < 2\epsilon + \sup_{a \in K} \left\| \widehat{\mathcal{G}}_N \circ \mathcal{F}_N \circ P_N(a_t) - \widehat{\mathcal{N}} \circ \mathcal{F}_N \circ P_N(a_t) \right\|_\infty \\
& \quad + \sup_{a \in K} \left\| \widehat{\mathcal{N}} \circ \mathcal{F}_N \circ P_N(a_t) - \widehat{\mathcal{N}} \circ \mathcal{N}_{FT}(a_t) \right\|_{L^2} \\
& \leq 2\epsilon + \sup_{\substack{\{\hat{w}(\xi)\} \in \mathbb{C}^{\mathcal{K}_N} \\ \|\hat{w}(\xi)\|_\infty \leq B'}} \left\| \widehat{\mathcal{G}}_N \left( \{\hat{w}(\xi)\} \right) - \widehat{\mathcal{N}} \left( \{\hat{w}(\xi)\} \right) \right\|_\infty \\
& \quad + \sup_{a \in K} \left\| \widehat{\mathcal{N}} \circ \mathcal{F}_N \circ P_N(a_t) - \widehat{\mathcal{N}} \circ \mathcal{N}_{FT}(a_t) \right\|_{L^2} \\
& < 3\epsilon + \sup_{a \in K} \left\| \widehat{\mathcal{N}} \circ \mathcal{F}_N \circ P_N(a_t) - \widehat{\mathcal{N}} \circ \mathcal{N}_{FT}(a_t) \right\|_{L^2} \\
& < 4\epsilon.
\end{aligned} \tag{9}$$

Since $t \in [0,T]$ is arbitrary, the above inequalities bring us to the desired result

$$\sup_{a \in K} \left\| \mathcal{G}(a) - \mathcal{N}(a) \right\|_{C\left( [0,T] \times H^{s'} \right)} < \epsilon.$$

# B  STABILITY

**Proof of stability**    Suppose $\sigma$ is a Lipschitz continuous activation function with Lipschitz constant $C$, and both $\sup_\xi \|R(t,\xi)\|_2^2$ and $\|W(t)\|_2^2$ are bounded by $M$ for every $t$. Let $\tilde{v}$ be a perturbed

initial condition with $\|\tilde{v} - v\|_2 < \epsilon$. Then by Plancherel theorem and Fourier convolution theorem, we have

$$
\begin{aligned}
&\|\mathcal{L}_\ell(t, \tilde{v}) - \mathcal{L}_\ell(t, v)\|_{L^2}^2 \\
&= \|\sigma(W_\ell(t)\tilde{v}(x) + (\mathcal{K}_\ell(t) * \tilde{v})(x)) - \sigma(W_\ell(t)v(x) + (\mathcal{K}_\ell(t) * v)(x))\|_{L^2}^2 \\
&\leq C\|W_\ell(t)(\tilde{v} - v)(x) + (\mathcal{K}_\ell(t) * (\tilde{v} - v))(x)\|_{L^2}^2 \\
&= C\int |W_\ell(t)(\tilde{v} - v)(x) + (\mathcal{K}_\ell(t) * (\tilde{v} - v))(x)|^2\, dx \\
&\leq C\int |W_\ell(t)(\tilde{v} - v)(x)|^2\, dx + C\int |R_\ell(t, \xi) \cdot \mathcal{F}(\tilde{v} - v)(\xi)|^2\, d\xi \\
&\leq C\int \|W_\ell(t)\|_2^2 |(\tilde{v} - v)(x)|^2\, dx + C\int \|R_\ell(t, \xi)\|_2^2 |\mathcal{F}(\tilde{v} - v)(\xi)|^2\, d\xi \\
&\leq CM\int |(\tilde{v} - v)(x)|^2\, dx + CM\int |\mathcal{F}(\tilde{v} - v)(\xi)|^2\, d\xi \\
&= 2CM\|\tilde{v} - v\|_{L^2}^2 \\
&< 2CM\epsilon.
\end{aligned}
$$

## C  EXPERIMENTAL DETAILS

In this section, we present our experimental settings in detail. All experiments were conducted on a single NVIDIA RTX 3090 GPU. We also detailed the algorithm of the proposed CTFNO in the one-dimensional case in Algorithms 1, 2, and 3. Algorithms can be extended to the higher dimensional case in a dimension-by-dimension manner.

---

**Algorithm 1** CTFNO (1-dim)

---

**Input:** input $x \in \mathbb{R}^{ch \times n}$ at initial $t = 0$ with grid resolution $n$, output timestep $t \in \mathbb{R}$, number of blocks $L$, width $w$, activation $\sigma$
1: $grid \leftarrow [0, 1/n, \ldots, (n-1)/n] \in \mathbb{R}^{1 \times n}$
2: $x \leftarrow \text{concat}(x, grid) \in \mathbb{R}^{(ch+1) \times n}$
3: $x \leftarrow \text{conv}_{1 \times 1}(x) \in \mathbb{R}^{w \times n}$
4: **for** $i = 1, 2, \ldots, L$ **do**
5: $\quad x \leftarrow \sigma(\text{SpectralFreqTimeConv1d}_i(x, t) + \text{TimeConv1d}_i(x, t)) \in \mathbb{R}^{w \times n}$
6: **end for**
7: $x \leftarrow \text{conv}_{1 \times 1}(x) \in \mathbb{R}^{ch \times n}$
8: **return** $x$

---

**Algorithm 2** SpectralFreqTimeConv1d

---

**Input:** input $x \in \mathbb{R}^{w \times n}$, time $t \in \mathbb{R}$, shared sinusoidal embedding $\phi : \mathbb{R} \to \mathbb{R}^c$, width $w$, Fourier modes $N$, number of heads $h$, learnable parameters $A \in \mathbb{R}^{h \times (c/h) \times 2N}$, $b \in \mathbb{R}^{1 \times 2N}$, $R \in \mathbb{C}^{w \times w \times N}$, Gershgorin stabilization constant $M$
1: $\hat{x} \leftarrow \text{FFT}(x)[\ldots, :N] \in \mathbb{R}^{w \times N}$ (filter $N$ Fourier modes after FFT)
2: $\phi(t) \leftarrow \text{reshape}(\phi(t)) \in \mathbb{R}^{h \times (c/h)}$ ('reshape' denotes a natural map $\mathbb{R}^c \to \mathbb{R}^{h \times (c/h)}$)
3: $\phi(t)(\eta, \cdot) \leftarrow \sum_k \phi(t)_{\eta,k} A_{\eta,k,\cdot} \in \mathbb{R}^{2N}$ for $\eta \in \{1, \ldots, h\}$
4: $\phi(t) \leftarrow \text{repeat}(\phi(t), w/h) + b \in \mathbb{R}^{w \times 2N} \simeq \mathbb{C}^{w \times N}$
5: $y \leftarrow \hat{x} \odot \phi(t) \in \mathbb{C}^{w \times N}$
6: $y(\cdot, \xi) \leftarrow \sum_j y_{j,\xi} R_{j,\cdot,\xi} \in \mathbb{C}^w$ for $\xi \in \{1, \ldots, N\}$
7: $norm(\cdot, \xi) \leftarrow \sum_j |\phi(t)|_{j,\xi} |R|_{j,\cdot,\xi} \in \mathbb{R}^w$ for $\xi \in \{1, \ldots, N\}$
8: $norm \leftarrow \max(norm/M - 1, 0) + 1$
9: $y \leftarrow \text{pad}(y/norm)$ ('pad' denotes zero-padding for frequencies $> N$)
10: **return** $\text{InverseFFT}(y)$

---

---

**Algorithm 3** TimeConv1d

---

**Input:** input $x \in \mathbb{R}^{w \times n}$, time $t \in \mathbb{R}$, shared sinusoidal embedding $\psi : \mathbb{R} \to \mathbb{R}^c$, width $w$, learnable
    parameters $B \in \mathbb{R}^{w \times w}$, Gershgorin stabilization constant $M$
  1: $\psi(t) \leftarrow FC(\psi(t), c, 2w) \in \mathbb{R}^{2w}$ (where $\psi(t) \in \mathbb{R}^c$)
  2: $W \leftarrow \psi(t)[\ldots, : w] \in \mathbb{R}^w, \ b \leftarrow \psi(t)[\ldots, w :] \in \mathbb{R}^w$
  3: $W \leftarrow B\mathrm{diag}(W) \in \mathbb{R}^{w \times w}$
  4: $y \leftarrow Wx \in \mathbb{R}^{w \times n}$
  5: $norm \leftarrow \sum_j |W|_{j,\cdot} \in \mathbb{R}^w$
  6: $norm \leftarrow \max(norm/M - 1, 0) + 1$
  7: $y \leftarrow y/norm + b$
  8: **return** $y$

---

### C.1 DATASETS

**Spiral** Data generation of Spiral, Stiff, Sawtooth, and Square follows the implementation of Holt et al. (2022) unless stated. We generate the Spiral dataset by following governing equation:

$$\dot{u}(t) = A \tanh u(t), \tag{10}$$

where $A = \begin{pmatrix} -1/8 & 1 \\ -1 & -1/8 \end{pmatrix}$. The other settings such as the number of training and test samples, initial value $u(0)$, and the number of timesteps follow the implementation of Holt et al. (2022).

**Stiff Van der Pol Oscillator** We generate data by following the instructions of (Van der Pol & Van Der Mark, 1927), which exhibits regions of high stiffness. The governing equation is

$$\dot{x} = y, \tag{11}$$
$$\dot{y} = \mu \left(1 - x^2\right) y - x, \tag{12}$$

where $\mu = 1000$. We sample initial conditions from $x(0) \in [0.1, 2], y(0) = 0$.

**Sawtooth & Square** We also explore the benchmarks on a periodic discontinuous function $u(t)$. We sample two datasets, namely sawtooth and square, and these are sampled by the equation $u(t) = \frac{t}{2\pi} - \lfloor \frac{t}{2\pi} \rfloor$, and $u(t) = 2 \left(1 - \lfloor 2 \left(\frac{t}{2\pi} - \lfloor \frac{t}{2\pi} \rfloor\right) \rfloor\right)$, respectively. We sample initial values $(t_0, u(t_0))$ by sampling $t_0$ in uniformly at random on the interval $[0, 2\pi]$. Each trajectories are generated from intervals of $[t_0, t_0 + 20]$.

**Reaction equation** The semi-linear ODE presented in Section 4.1 has an analytic solution

$$u(t, x) = \frac{f(x) e^{\rho t}}{f(x) e^{\rho t} + 1 - f(x)}, \tag{13}$$

for an initial condition $f(x)$. The reaction coefficient $\rho$ is chosen to be 6 and the domain is unit interval. Initial conditions randomly generated as $\frac{1}{2} \left(z_1 \sin(2\pi k_1 x) + z_2 \sin(2\pi k_2 x)\right) + z_3 e^{-x} + 2$, where $z_i \sim N(0, 1)$ and $k_i$ is a uniformly sampled integer. Analytic solutions are attained up to $t = 1$. The solution data is generated by solving (13). The spatial grid is discretized with resolution of 100 along time step size of $\Delta t = 0.02$.

**Heat equation** The one-dimensional heat equation used in Section 4.1 takes the form

$$\begin{cases} \frac{\partial u}{\partial t} = \nu \frac{\partial^2 u}{\partial x^2}, & x \in (0, 1), \ t \in (0, 2.5] \\ u(0, x) = u_0(x), & x \in (0, 1), \end{cases} \tag{14}$$

with periodic boundary conditions. It describes how a quantity such as heat diffuses through a given region over time. The initial function $u_0(x)$ is generated from the Gaussian random field $\mathcal{N}\left(0, 20^2 \left(-\triangle + 3.5^2\right)^{-2.5}\right)$, where $\triangle$ refers to the Laplacian. The diffusivity constant $\nu$ is set to be 0.001 and the solution data is generated by exactly solving (14) in Fourier space on a uniform spatial grid with resolution 1024 along time step size $\triangle t = 0.05$.

**Burgers' equation** The one-dimensional Burgers' equation with diffusive regularization is a canonical non-linear PDE, taking the form

$$\begin{cases} \frac{\partial u}{\partial t} + u\frac{\partial u}{\partial x} = \nu\frac{\partial^2 u}{\partial x^2}, & x \in (0,1), \ t \in (0,1] \\ u(0,x) = u_0(x), & x \in (0,1), \end{cases} \tag{15}$$

with periodic boundary condition. It is a dissipative nonlinear system with shock formation and has various applications including the flow of viscous fluid dynamics. The viscosity is set to $\nu = 0.001$. Starting from initial functions sampled from $\mathcal{N}\left(0, 7^2\left(-\triangle + 49\right)^{-2.5}\right)$, numerical ground truth is generated separately where the linear diffusion component is exactly solved and the remaining nonlinear part is solved in Fourier space using forward Euler method as in (Li et al., 2021). The spatial domain is discretized with resolution 1024. The training set consists of 400 periodic trajectories and each trajectory has regularly-sampled time points with $\triangle t = 0.005$ and we test the trained models for 100 trajectories.

**MuJoCo** We use a trajectory of physical simulation for the hopper with three joints and four body parts. Each time series is 14-dimensional, consisting of a five-dimensional position, six-dimensional velocity, and three-dimensional action. Rubanova et al. (2019) generated $10,000$ sequences of 200 timesteps and used 80% of the data for training and the rest 20% for evaluation. For the interpolation task, they randomly sampled 100 consecutive observation timesteps. For the prediction task, they use all 200 observation timesteps and divide them into two parts: the first 1/3 as an input for the model and the latter 2/3 as an output to be forecasted. For all the tasks, they randomly sampled a portion of the input timesteps at a specified rate and masked the rest of them. Note that the values used for the subsampling ratio are 10%, 20%, 30%, and 50%.

**PhysioNet** This is a publicly available dataset of 8000 time series describing the stay of patients within an ICU over 48 hours. For each patient, 41 biomedical features are irregularly observed and converted to one minutely resolution. In (Rubanova et al., 2019), they used 80% of the data for training, and the rest 20% for evaluation, as in MuJoCo. For the interpolation task, they did not subsample the data because the measurements were already sparse. For the prediction task, they halved the data so that the first 1/2 timesteps were used for inputs, and the latter 1/2 timesteps were used for outputs to be forecasted. Note that Rubanova et al. (2019) also performed per-sequence classification experiments on PhysioNet. Since Latent ODEs with RNN or ODE-RNN encoders compress the whole input data to a single latent vector $z_0$ and an MLP classifier receives only $z_0$ as its input, such a per-sequence classification does not depend on all the other latent vectors $z_1, \ldots, z_T$. This is inconsistent with the goal of CTFNO, which focuses on learning representations of the inherent dynamics for all timesteps. Hence we excluded per-sequence classification experiments in this paper.

**Activity** This dataset consists of time series from five individuals performing serveral activities (i.e. walking, standing, laying, etc). Each time series includes 12 features indicating tags attached to their belt, chest and ankles. We used the same pre-processing as (Rubanova et al., 2019), resulting in 6554 sequences of 211 time points. The task is to classify each time point into one of seven types of activities and we performed per-time-point classification experiments as in (Rubanova et al., 2019).

**Plane Vibration** The dataset is consisted of time 0 to 73627, with five attributes recorded per timestamp. We randomly take out 10% of data to make the time series irregularly sampled. Following the implementation of (Xia et al., 2021), we use the first 50% of data as our train set, the next 25% as a validation set, and the rest as a test set. We divide each set into partitions of consecutive 64 timestamps of the irregularly-sampled time series, and our goal is to forecast 8 consecutive timestamps starting from the last timestamp of the segment.

## C.2  NETWORK STRUCTURE

**Time Embedding** In order to obtain $\varphi(t)$ and $\phi(t)$, we first adopt a positional embedding of transformer (Vaswani et al., 2017). A time encoding function converts an one-dimensional time into a multi-dimensional input by passing the time $t$ through trigonometric functions of varying frequencies:

$$t \mapsto (\sin(\omega_i t), \cos(\omega_i t)), \ \omega_i = 10^{-\frac{4i}{m}},$$

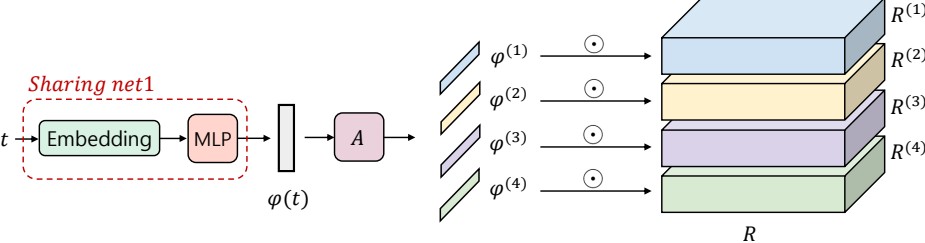

Figure 4: The architecture of multi-head CTFNO.

where $i$ runs over a range of integers $\{0, \ldots, m-1\}$ with encoding dimension $m$. Then, we pass MLP layers with the number of hidden dimension $c$. The contextual visualization of sharing net $\varphi(t)$ can be found in Figure 4.

**Fourier kernel**    Time modulation multiples a single value $\varphi_\ell(t, \xi) \in \mathbb{R}$ to a $d_v \times d_v$ Fourier kernel $R_\ell(\xi)$, that is,

$$R_\ell(t, \xi) = \varphi_\ell(t, \xi) \cdot R_\ell(\xi).$$

However, often there are multiple different aspects channel elements attend to, and scalar multiplication may not be a good option for it. To attempt to lift this restriction, we employ parallel heads. Specifically, we divide the Fourier kernel $R_\ell$ into $h$ kernel blocks $R_\ell^{(i)}(\xi) \in \mathbb{R}^{d_k \times d_v}$ for $i = 1 \ldots, h$, where $h$ is the number of heads and $d_k = d_v/h$. Afterward, we concatenate the heads and combine them with a final kernel matrix. (See Figure 4.) This operation can be expressed as:

$$R_\ell(t, \xi) = \text{Concat}\left(R_\ell^{(1)}, \ldots, R_\ell^{(h)}\right),$$

$$R_\ell^{(i)}(t, \xi) = \varphi_\ell^{(i)}(t, \xi) \cdot R_\ell^{(i)}(\xi), \quad i = 1, \ldots, h.$$

The multi-head kernel allows the model to share different representations at different channels.

**Model architecture for 2D data**    While the Fourier kernel $R$ plays a key role in FNO models for learning the solution operator of PDEs, it requires quite many parameters ($d_v^2$ for each frequency). This parameter redundancy worsens for two-dimensional data; images in Section 4.3. To overcome parameter inefficiency, we modify the Fourier kernel as follows. One modification is the use of block-diagonal structure of $R$, as proposed in (Guibas et al., 2021). This reduces the number of parameters, however, the diagonal structure does not fully mix the information between channels, which may lessen the expressive power of the model. To compensate for this, we additionally introduce a local $1 \times 1$ convolution layer $V$ in Fourier space. The overall architecture can be written as

$$\mathcal{L}_\ell(v)(x) = \sigma\left(W_\ell v(x) + \mathcal{F}^{-1}\left(R_\ell(\xi) \cdot \mathcal{F}(v)(\xi) + V_\ell \mathcal{F}(v)(\xi)\right)(x)\right). \tag{16}$$

We empirically show that this improves the training ability of FNO with significantly less parameters.

**Lifting Layers for Real Application**    Fourier transform may be hard to embody discontinuous functions because Fourier filters are global sinusoidal functions. As a consequence, FNO has difficulty in learning sharp features (Lu et al., 2022). In many applications, data is discontinuous, such as images. Hence, the performance of FNO models is plagued by the discontinuous nature of such input data. Contrary to regular PDE problems where the pointwise lifting is adequate, we hereby propose an alternative lifting layer to provide a smoother form of the data. A low pass filter is the standard way of smoothing out the data; it tends to retain the low frequency modes while truncating out the high frequency information. To cutoff high spatial frequency, we additionally introduce a low pass filter after the original lifting layer. For parameter efficiency, the low pass filter is implemented by an $1 \times 1$ convolution layer in Fourier space, as we introduced in the preceding paragraph. In this way, input data is smoothed by decreasing the disparity between neighboring values without loss of information.

Table 7: The number of parameters of each model in synthetic experiments. Low indicates the experiments on Spiral, Stiff, Sawtooth, and Square. High denotes the high dimensional experiments, including Heat and Burgers' equations. LODE, NF, NL, DON, and PDN stand for Latent ODE, Neural Flow, Neural Laplace, DeepONet, and POD-DeepONet, respectively.

| Data | NODE | ANODE | LODE | NF | NL | DON | PDN | CTFNO |
|------|------|-------|------|-----|-----|-----|-----|-------|
| Low | 17025 | 17282 | 18565 | 18307 | 17194 | - | - | 17858 |
| High | 5.24M | 5.45M | - | 16.57M | 4.51M | 1.58M | 1.78M | 2.38M |

## C.3 HYPERPARAMETERS AND OTHER EXPERIMENTAL DETAILS

We choose the hyperparameters and other experimental details that delivered the best performance for the baseline models. Note that each model has different optimal optimizer settings, so we carefully choose the learning rate and its decay rate that show the best performance.

### C.3.1 SYNTHETIC

**Low dimensional data** Low dimensional data includes Spiral, Stiff, Sawtooth, and Square. All the experiments on these datasets are trained for 1000 epochs with a batch size of 120. We do not normalize data since all data are properly bounded. Implementation of NODE, ANODE, Latent ODE, Neural Flow, and Neural Laplace follow (Holt et al., 2022), other than the data normalization. For CTFNO, a single data point is linearly transformed into a 16-dimensional vector. Then, we regard the temporal dimension of 100 as a channel size and convert it to 16 by a lifting layer. We use the learning rate of 1e-3 with no decay. We do not use the stabilization constant and use Gaussian error Linear Units (GeLU) (Hendrycks & Gimpel, 2016) activation in all FNO-based experiments. In the time-embedding layer, we employ Sigmoid Linear Units (SiLU) (Elfwing et al., 2018) activation function. Note that all experiments for CTFNO use the same activation function.

**High-dimensional data** In Reaction ODE, Heat, and Burgers' experiments, which are high dimensional datasets, we set comparison models to have a large number of parameters, at least 1M. In NODE and ANODE experiments, we set an ODE function $f(t, u(t))$ to a five layer Multi-Layer Perceptron (MLP), of 1024 units. For ANODE, we set the augmented dimension of zeros to 100. We employ the learning rate of 1e-5, and the number of epochs of 2000. Other hyperparameters follow the NODE implementation of Holt et al. (2022). Moreover, in the implementation of Neural Laplace, we set the latent dimension and the number of hidden units to 64. Other hyperparameters are set the same as low-dimensional data experiments. In the implementation of Neural Flow, we use a 32-layered coupling flow. Each coupling flow has two hidden layers with 128 hidden dimensions. Timesteps are embedded into 32 sinusoidal features. We set the learning rate of 1e-3 with the decay rate of 0.8 for every 200 epochs. Other hyperparameters for Neural Flow follow the synthetic experiments in (Biloš et al., 2021). In DON and PDN experiments, we use the number of channels of 256 with six residual layers for each branch and trunk net. We use SiLU activation, the training epochs of 20000, the batch size of 50, and the learning rate of 1e-3. The hyperparameters of FNO-based models, including CTFNO, are summarized in Table 8. We train FNO-based models with a batch size of 20, the training epochs of 2000, and the learning rate of 1e-3 with a decay rate of 0.8 for every 100 epochs.

### C.3.2 REAL TIME-SERIES

Our implementation builds on open-source codes [1][2][3][4] (MIT License). For all real time-series datasets, we refer to the results of RNN-VAE, ODE-RNN and Latent ODEs reported in (Rubanova et al., 2019). For CTFNO, we commonly use the number of heads of one and two padding dimensions. Note that all layers are equipped with GeLU activation, except for only time-embedding layers with SiLU activation. Other model hyperparameters for CTFNO are shown in Table 9. Moreover, we use the batch size of 50, and the Adamax optimizer (Kingma & Ba, 2014) with the learning rate 1e-2.

---

[1] https://github.com/rtqichen/torchdiffeq
[2] https://github.com/YuliaRubanova/latent_ode
[3] https://github.com/mbilos/stribor
[4] https://github.com/hedixia/HeavyBallNODE

Table 8: Hyperparameter settings of CTFNO, FNO2d, and FNOseq on synthetic data. Low indicates the experiments on Spiral, Stiff, Sawtooth, and Square data. Layers, Modes, and Channels are the number of Fourier layers, Fourier modes, and channels, respectively. The first value in Time channels is the dimension used for time embedding and the second value is the dimension of sinusoidal embedding of time. Params denotes the size of model parameters.

| Model | Data | Layers | Modes | Channels | Time channels | Params |
|---|---|---|---|---|---|---|
| CTFNO | Low | 3 | 4 | 16 | (32, 16) | 17858 |
| | Reaction | 2 | 32 | 64 | (512, 128) | 1.82M |
| | PDEs | 2 | 64 | 64 | (512, 128) | 2.38M |
| FNO2d | Reaction | 3 | (32, 16) | 32 | - | 7.00M |
| | PDEs | 3 | (64, 16) | 32 | - | 12.00M |
| FNOseq | PDEs | 4 | 64 | 64 | - | 2.02M |

Table 9: Hyperparameter settings of CTFNO on real-world time-series data. Layers, Modes, and Channels are the number of Fourier layers, Fourier modes, and channels, respectively. The first value in Time channels is the dimension used for time embedding and the second value is the dimension of sinusoidal embedding of time. $M$ stands for the stabilization constant. Params denotes the number of model parameters.

| Data | Layers | Modes | Channels | Time channels | $M$ | Params |
|---|---|---|---|---|---|---|
| MuJoCo | 3 | 10 | 32 | $(64, 32)$ | 10 | 150K |
| PhysioNet | 1 | 10 | 32 | $(64, 8)$ | - | 68K |
| Activity | 2 | 10 | 32 | $(64, 32)$ | 10 | 124K |
| Plane Vibration | 2 | 10 | 8 | $(16, 8)$ | 1.2 | 6K |

**Latent ODE framework**  We use the same Latent ODE framework proposed in (Rubanova et al., 2019). First, a given input time series is passed through an RNN encoder. Here, a latent vector $z_0$ is sampled using the last feature vector as the mean $\mu$ and standard deviation $\sigma$, which is the same as VAE (Kingma & Welling, 2014; Szegedy et al., 2013). Now, assuming an implicit dynamics in the latent space with $z_0$ as the initial point, each model (such as Neural ODEs and CTFNO) returns an output vector at the desired timesteps. Finally, after passing through a decoder with a fully connected layer, the difference between the output and the target becomes the loss function for training.

**MuJoCo**  Overall experimental settings follow (Rubanova et al., 2019), unless stated. We use the latent dimension of 20, the hidden state dimension of 100 for the RNN encoder, and the training epochs of 300. Since Biloš et al. (2021) did not contain the results of interpolation and prediction tasks with different subsampling ratios, we carefully trained Neural Flow, following the official implementation[5]. Following (Rubanova et al., 2019), we employ the importance weighted likelihood loss (Burda et al., 2015).

**PhysioNet**  Overall experimental settings follow (Rubanova et al., 2019), unless stated. We use the latent dimension of 20 and the hidden state dimension of 40 for the RNN encoder. The training epochs are 200 and 50 for the interpolation task and the prediction task, respectively. As in (Rubanova et al., 2019), we employ the importance weighted likelihood loss (Burda et al., 2015) without leveraging classification loss.

**Activity**  Overall experimental settings follow (Rubanova et al., 2019), unless stated. We use the latent dimension of 20, the hidden state dimension of 100 for the RNN encoder, and the training epochs of 200. For classification, the cross entropy loss was used, as in (Rubanova et al., 2019).

---

[5]`https://github.com/mbilos/neural-flows-experiments`

Table 10: The number of parameters for each models for image classification.

| Data | CNN | ANODE | FNO |
|---|---|---|---|
| MNIST | 78K | 83K | 66K |
| CIFAR10 | 154K | 168K | 153K |

### C.3.3 Studies on stability

**Plane Vibration**   In this experiment, we follow the settings of Xia et al. (2021). For CTFNO, five-dimensional attributes are concatenated with input time and then embedded into 20-dimensional space by a linear transformation. We regard temporal dimension 64 as a channel size and convert it to eight by a lifting layer. We train the model for 500 epochs and use the learning rate of 1e-2 with decay rate 0.5 per 100 epochs. We trained and evaluated with MSE loss.

**Image classification**   All of the experiments use gradient clipping of 10 and cross-entropy loss. We use a learning rate of 1e-3 and weight decay of 1e-4 for ANODE and CNN. Other implementation details of ANODE follow Dupont et al. (2019). We use five convolutional layers in CNN experiments. For MNIST, we apply a kernel size of seven for the first two convolutional layers and 32 number of channels. For CIFAR10, we use a kernel size of seven for the first convolutional layers and 64 number of channels. Other convolutional layers have a kernel size of three. We train CNN for 30 epochs. Note that accuracy of both ANODE and CNN tend to decrease after the designated number of epochs.

In FNO experiments, FNO with weight decay (WD) and FNO with normalization (Stab), we train models for 100 epochs with the learning rate of 5e-3. We used three Fourier layers with six Fourier modes. For MNIST, we employ the number of channel of 32, and three-dimensional zero padding. For CIFAR10, we employ the number of channel of 64, and four-dimensional zero padding. For the experiments with decay, we choose the decay rates 1e-6 and 1e-4 for MNIST and CIFAR10, respectively. For the experiments with normalization, we employ the spectral norm of 1.2 for the normalization bounds. Note that GeLU activation function was used for both CNN and FNO experiments.

To estimate the robustness of the trained model, we consider three commonly-used perturbation schemes, namely random Gaussian perturbations, FGSM, and PGD attacks. For random Gaussian attack, we use zero mean Gaussian noise with standard deviation of 50/255, 100/255 for MNIST and 15/255, 20/255 for CIFAR10. For adversarial attacks, we use 30/255, 50/255 attack for MNIST and 2/255, 5/255 for CIFAR10. The number of steps of PGD attack is 10.

## D   Further Results

### D.1   Comparison to various time-injected FNO-based models

In Section 3.2, we construct various baselines injecting time $t$ into the network for comparative analysis of CTFNO. This section includes implementation details. Since CTFNO puts $t$ into weights of the Fourier layers, the other way to inject $t$ is to join it in the input function $u_0(x)$ or intermediate features $v_0(x), v_1(x), \ldots, v_L(x)$. To cover variants of such cases, we constructed three other baselines adding $t$ into the input or feature space as discussed in Section 3.2. The architecture of these three scenarios is pictorized in Figure 5.

Comparison results of these three baselines together with FNO2d (proposed in the original FNO paper) and CTFNO in ODE/PDE problems are reported in Table 11. The results show that our model significantly outperforms other models. The results further show that three scenarios, which are structures that can continuously handle arbitrary time $t$, are similar to or worse than FNO2d that can only evaluate the solution value at times on a specific grid. Even though all three baselines and CTFNO model continuous time operators, it was found through the experiments which only the structure of CTFNO is effective. The difference from the baselines is that CTFNO inserts temporal information into weights. The rationale of the way of imposing temporal information on CTFNO is based on Green's formula of time-dependent PDEs. The results validate that it is much more effective to construct a model structure based on the theoretical Green's formula to approximate the

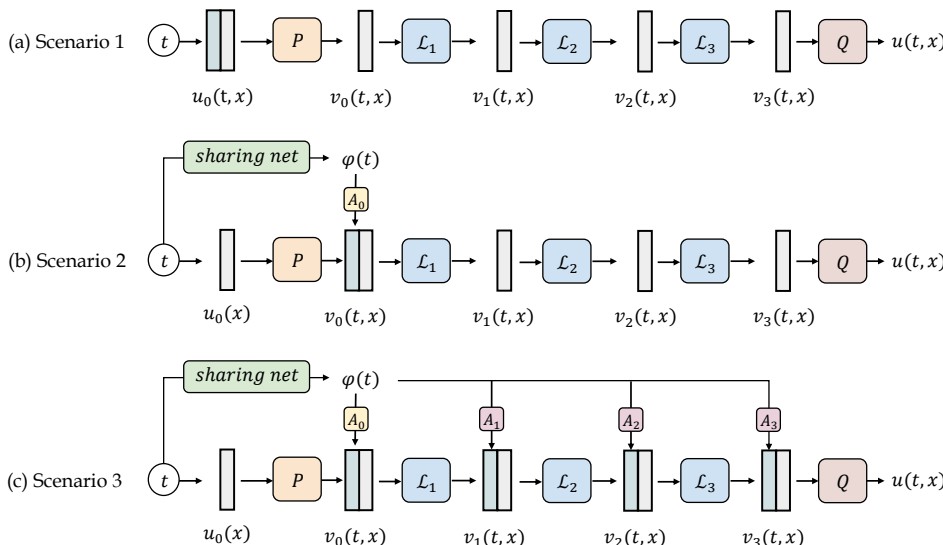

Figure 5: The visualization of architecture of three baselines: (a) Concatenates the input time $t$ to an input function. (b) Concatenates the encoded time $\varphi(t)$ to the lifted input function. (c) Concatenates the encoded time to the lifted input function and intermediate features. Here, all $\boxed{A}$ denotes a learnable affine transform.

solution operator of PDE. In summary, we validate through additional experiments that the structure of CTFNO designed based on the solution operator of the PDE (what we want to learn) is very effective in actually learning the PDE.

Table 11: Results of time-aware FNO models in Synthetic Data

| Relative $L^2$ ($\times 10^{-2}$) | Reaction ODE | Heat | Burgers |
|---|---|---|---|
| Scenario 1 | 0.19 | 11.60 | 19.6 |
| Scenario 2 | 0.13 | 2.84 | 18.6 |
| Scenario 3 | 0.14 | 2.68 | 18.9 |
| FNO2d | 0.16 | 0.58 | 10.32 |
| CTFNO | **0.04** | **0.22** | **5.68** |

### D.2 STABILITY

As we discussed in Section 3.4, the upper bound $M$ on the $\|R(t,\cdot)\|_2$ and $\|W(t,\cdot)\|_2$ emanates from the necessity of stabilizing the proposed model. In Section 4.3, we conducted experiments to evaluate the robustness of our proposed stabilization scheme, and compare it with the vanilla FNO and weight decay. From Tables 5 and 6, FNO trained with stabilization outperforms vanilla FNO on all types of perturbations.

$M$ is a user-defined hyperparameter. Here, we analyze the performance of FNO to illustrate how increasing stabilization parameter $M$ leads to a more robust model. Experiments are carried out on CIFAR10 and we perturb images by PGD-5/255 attack. The left panel of Figure 6 shows the accuracy and robustness of the learned models for several values of $M$. We can see that imposing stability does come with trade-offs between test accuracy and robustness on the attack. The accuracy is higher at low $M$ and it becomes decreasing as $M$ increases, which makes the model more robust on various attacks. Moreover, the models with weak stabilization show brittle training procedures. Also, the model degrades the overall performance as stabilization effects become dominant. Results certify that small corruptions or extra noise in the input are not likely to change the output of the network with proper stabilization.

Moreover, the stability of the model is related to how well the model generalizes on the data on which it has not trained, which in turn is related to overfitting. This can be confirmed in Figure 6 (right).

It presents the evolution of the loss function of models with and without stabilization. As one can see, after some iterations, the test accuracy of the model without stabilization has started to decrease. This means that the longer we train the model, the more specialized the weights will become to the training data. This is evidence of overfitting. On the other hand, the model with stabilization keeps increasing its test accuracy. This confirms that the stabilization prevents overfitting and helps the model to work better on unseen data.

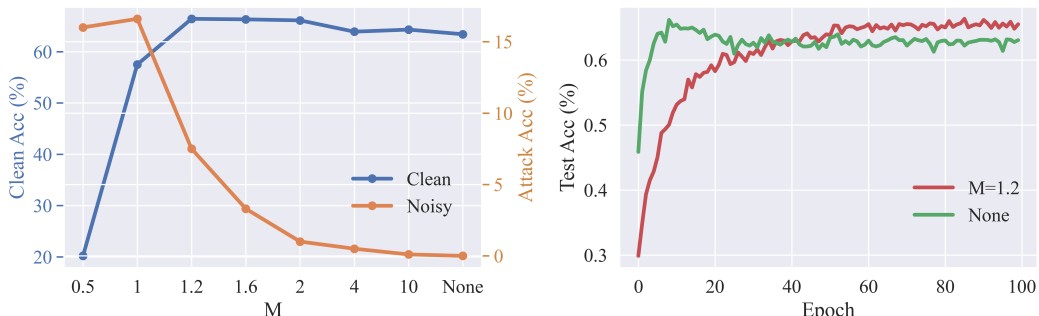

Figure 6: The Trade-off between the accuracy and robustness on PGD-5/255 attack (left) and observation for overfitting (right). Tests are conducted on CIFAR10 (§4.3).

The relation between the stabilization and overfitting was also discussed in Section 4.3 on the Plane Vibration dataset. By increasing $M$, the training MSE loss gradually decreases, while the test MSE eventually goes up (see Figure 3). Figure 7 plots the distribution of the $L^2$ norm of weights of trained CTFNO with $M = 1.2$ and 10 on the plane vibration data. The weights of the model trained with $M = 10$ have larger $L^2$ norms. As we analyzed in Theorem 3.2, the weights that have a large spectrum make the network unstable. More precisely, the spectrum of the weights grows in size to handle the specifics of the training data. As the weights become specialized to the training data, overfitting occurs. On the other hand, the trained model with $M = 1.2$ learned weights with a spectrum smaller than 1. It forces the network to have small changes in output for small changes in the inputs, which gives more ability to generalize better.

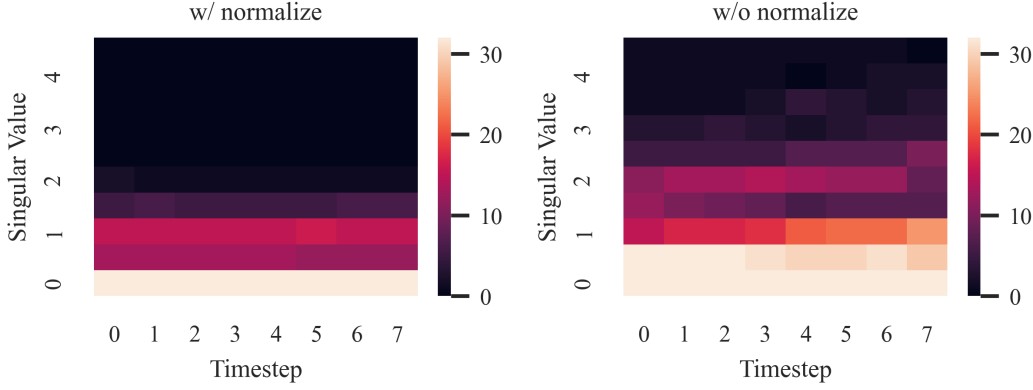

Figure 7: Plot of the singular values of weights of CTFNO with (left) and without (right) stabilization on the plane vibration dataset (§4.3).

## D.3 GENERALIZATION TO RESOLUTION

Classical neural networks that map between finite-dimensional Euclidean spaces are grid-dependent, and thus cannot generalize over image resolution. By viewing images as functions, FNO is not tied to a specific discretization and can be applied to arbitrary discretization

Table 12: Resolution invariance of FNO.

| Dataset | Original | Resolution | | |
|---|---|---|---|---|
| | | ×2 | ×4 | ×8 |
| MNIST | 98.8 | 98.6 | 98.6 | 98.6 |
| CIFAR10 | 65.5 | 64.8 | 64.6 | 64.7 |

of $a(\mathbf{x})$. To validate the resolution invariance of FNO, we classify high-resolution images using FNO trained on the original low-resolution images. Table 12 validates that FNO is able to learn from coarse images and generalize to higher resolutions.

## D.4 SPEED IMPROVEMENTS

Table 13: Measured training and inference time

| Model | Training (second/epoch) | Inference (second/epoch) |
|---|---|---|
| Neural ODE | 23.6 | 2.20 |
| Neural Flow | 4.38 | 0.62 |
| CTFNO | 7.39 | 0.89 |

We carefully measured execution time on the MuJoCo experiment (§4.2) and the results are reported in Table 13. The main reason for this difference is that numerical integration used in neural ODEs is computationally expensive, and Neural Flow and CTFNO are neural operators that directly represent the solution trajectory of ODE and PDE, respectively. In other words, the neural operator's replacement of the role of ODE solvers has contributed significantly to overcoming the increased complexity of PDEs compared to ODEs and reducing time costs.

## D.5 ADDITIONAL SYNTHETIC RESULTS

We investigate the generalization ability of our method to time points beyond those that were used for training on the two-dimensional spiral data in Section 4.1. Predicted and extrapolated trajectories are depicted in Figure 8 together with the ground truth. Each model is trained to predict spiral trajectories at 100 time points (blue lines in Figure 8). And we run trained models to forecast 500 future time steps (red lines in Figure 8). We can observe that our CTFNO is better at generalizing for extrapolation compared to baseline models. Our CTFNO correctly extrapolates the spiral trajectory for beyond the training time interval, converging to the equilibrium of the spiral. From Table 2, we have seen that CTFNO best approximates the spiral comparing against the benchmarks, achieving the lowest RMSE, The results depicted in Figure 8 confirms that CTFNO accurately captures the dynamics of the spiral during training and predicts well even for non-training times, based on memories analyzed in the past.

We also include additional heatmaps for reaction, heat, and Burgers' equations (§4.1). See Figure 9, 10, and 11.

## E BROADER IMPACT

We introduced a new framework for modeling temporal dynamics of observed data. It has a wide range of potential applications, some of which we investigated in this paper. We explored a healthcare dataset and here we hope to bring affirmative impact in medical applications. We also examined the modeling the vibration of an airplane and we expect it to contribute to the advancement of airplane designs and civil engineering. Furthermore, as many problems have arisen in sciences and engineering tied with complex PDE systems, we expect that our work has potential applicability in the enormous area such as climate forecasting, epidemics, molecular simulations, micro-mechanics, and modeling turbulent flows. PDEs can also be applied in the development of military equipment. As with all numerical methods, however, it is not a work of developing a technique to go to warfare, and we hope and encourage users of our model to concenter on the positive impact of this work.

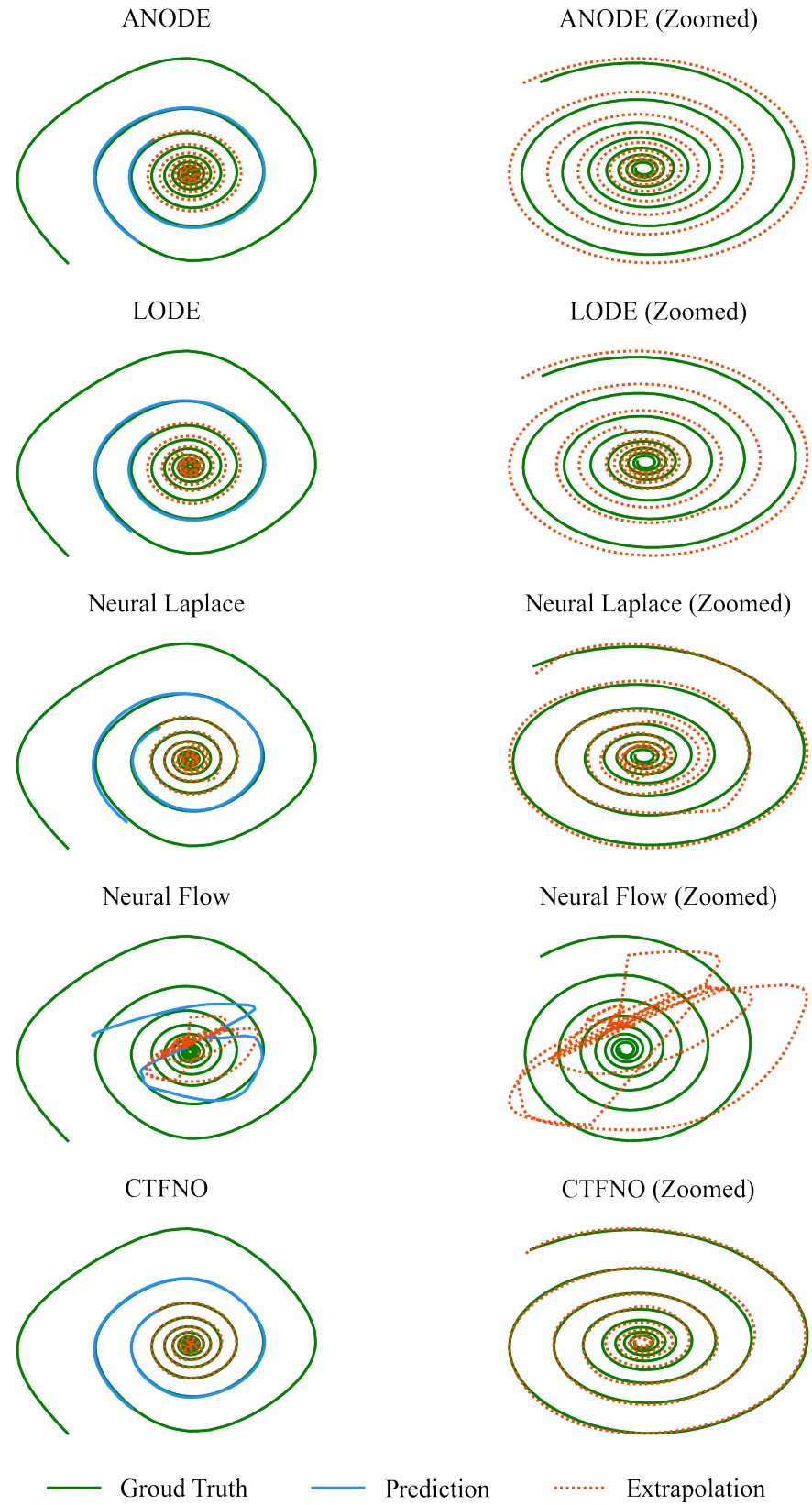

Figure 8: (Left) Prediction and extrapolation for beyond the training time points of the two-dimensional spiral (§4.1), and (Right) the zoomed extrapolation trajectories with a comparison to ground truth. CTFNO performs best not only for prediction but also for extrapolation.

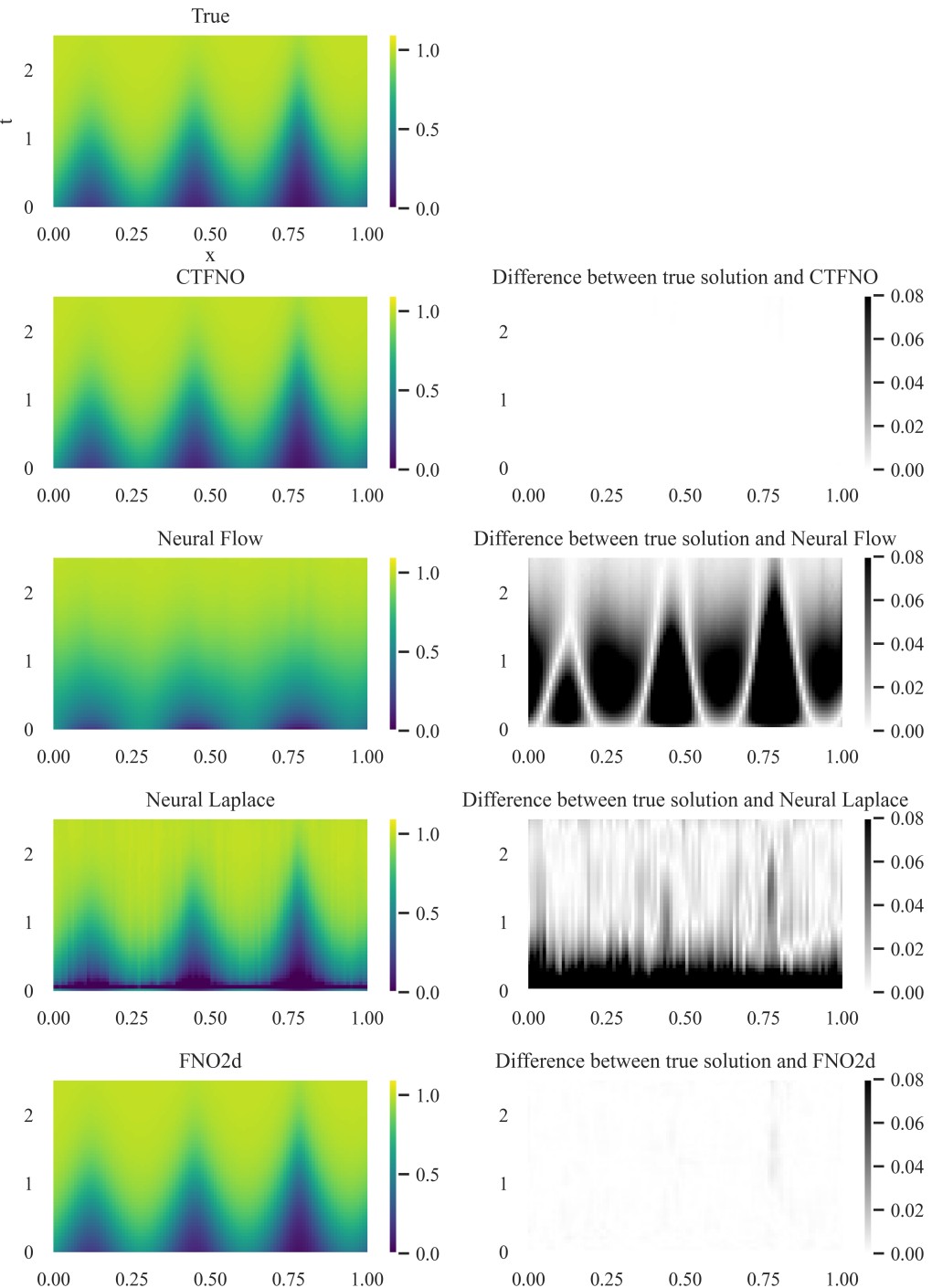

Figure 9: Heatmap of exact and predicted solution for reaction ODE (§4.1). This shows that our model can exactly represent the dynamics of the reaction ODE.

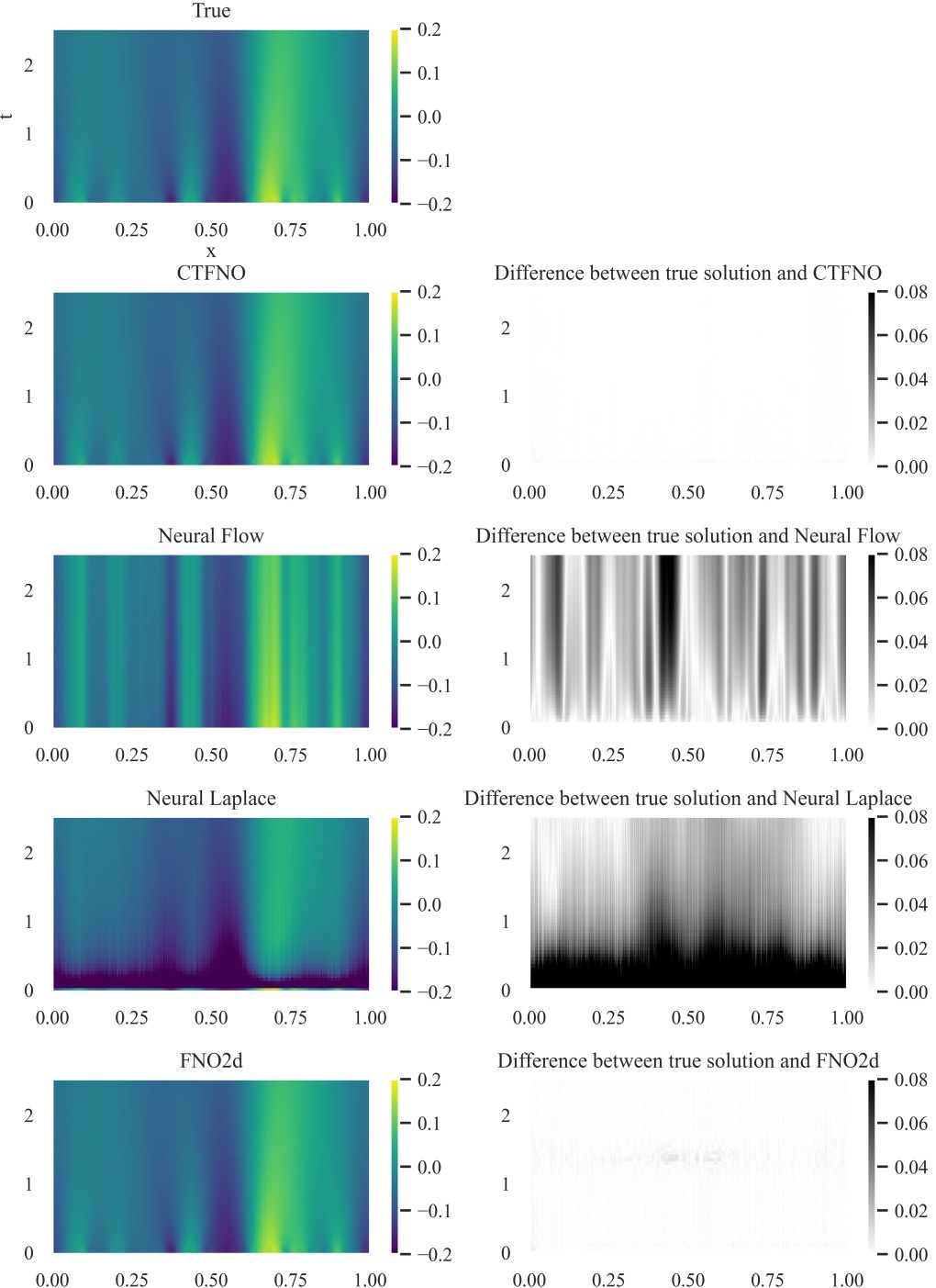

Figure 10: Heatmap of exact and predicted solution for heat equation (§4.1).

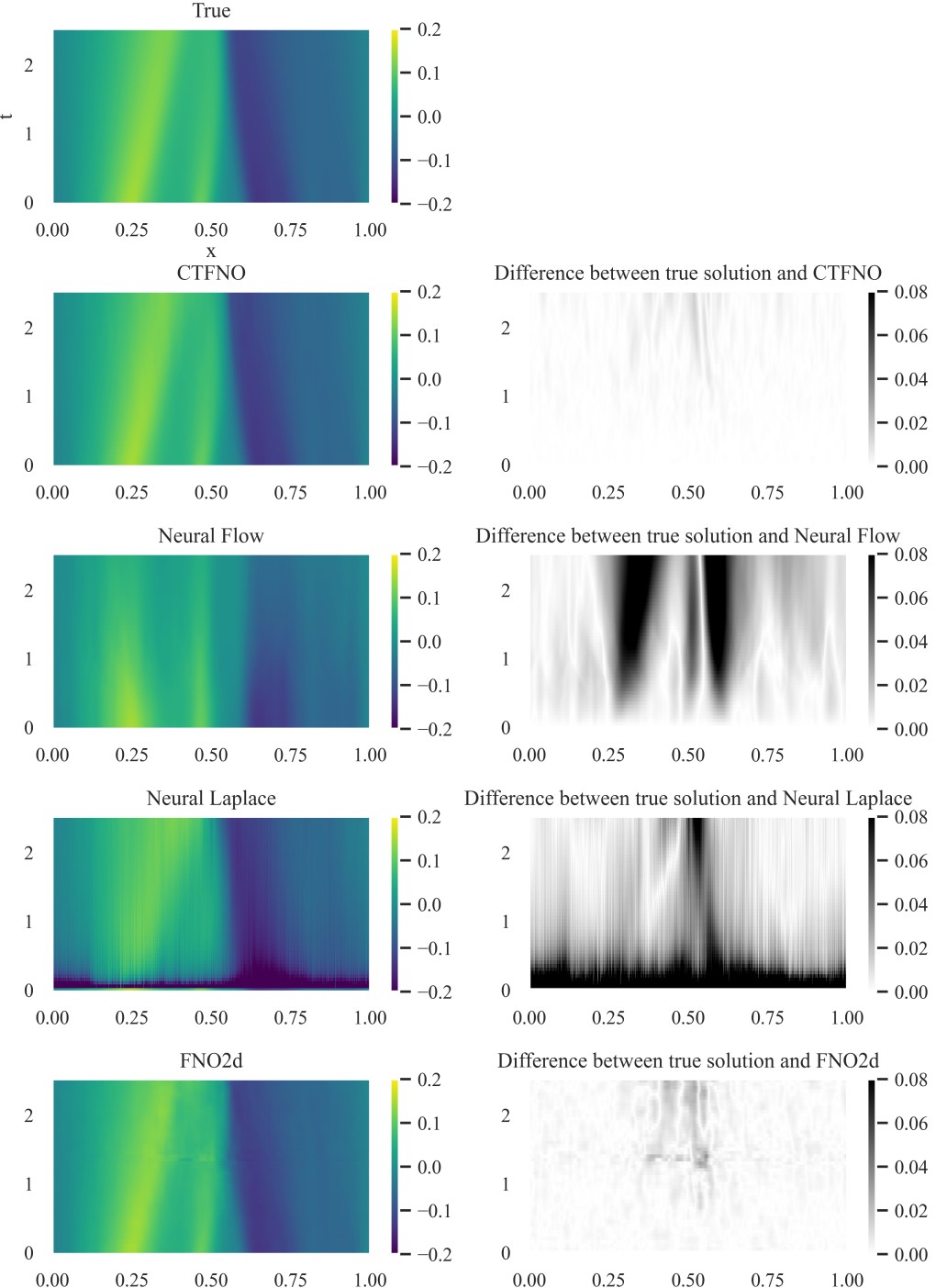

Figure 11: Heatmap of exact and predicted solution for Burgers' equation (§4.1).

