# OpenReview forum: "Learning PDE Solution Operator for Continuous Modeling of Time-Series"
_ICLR.cc/2023/Conference — Submitted to ICLR 2023_

### Official Review · Reviewer_dm3F · 2022-10-16

**Confidence:** 4
**Correctness:** 3
**Technical Novelty And Significance:** 3
**Empirical Novelty And Significance:** 2
**Recommendation:** 6

**Clarity, Quality, Novelty And Reproducibility:**

The paper is written relatively clear, although a red line is missing a bit. There is a lot emphasis on stability and the Gershgin discs normalization, discussion of which is missing in the experiments. I still missing the real intuition why the MuJoCo experiments are done. Altogether there seems to be enough novel and important ideas. In the current version, the experiments are not reproducible, important information and description is missing,

**Strength And Weaknesses:**

Strength:
The paper addresses an important topic, namely that many neural PDE surrogates are trained on fixed time grids. Whereas DeepONets like operator learning methods have time dependencies naturally encoded via the trunk net, Fourier Neural Operators do not.

The Gershgorin disc normalization is an interesting way of enforcing stability.


Weaknesses:
Under the hood, the newly proposed architecture conditions on the time variable. Conditioning is a super important property in various computer vision tasks and natural language translation tasks. As such a lot of different techniques have been developed:  for example, the encoding of the input into a vector representation by using sinusoidal Fourier embeddings as is common in Transformers, or AdaGN (Nichol et al., 2021), which is based on affine transformations of normalization layers via projections of the embeddings. I am not fully convinced with the three presented baseline approaches. They read rather week, but ok, it might just be really hard to condition for FNOs. A different why of encoding parameters in the Fourier space can be found e.g. in Poli et al. Potentially also relevant is https://openreview.net/forum?id=Uk40pC45YJG

Experiment-wise I am missing how the new time modulation improves stability. I would have liked to see long rollouts, and the conditioning of different time windows.  In that sense I have also waited for studies which address the effect of the Gershgorin disc normalization. That is missing in the current version and makes me wonder what the whole theoretical reasoning is about.

Also experiment-wise for e.g. heat equation and Burgers’ equation it is not 100% clear to me what is the input to the neural network surrogate and what is the output, and where the conditioning on time comes into the game. I would have expected curves showing errors for e.g. 100 step rollouts, 200 step rollouts, and so on. Also I am missing how FNO2d is used (input and output?)  I am also missing parameter and runtime comparisons. All in all that makes it really hard to judge the experiments.

Nichol, Alexander Quinn, and Prafulla Dhariwal. "Improved denoising diffusion probabilistic models." International Conference on Machine Learning. PMLR, 2021.
Poli, Michael, et al. "Transform Once: Efficient Operator Learning in Frequency Domain." ICML 2022 2nd AI for Science Workshop. 2022


**Summary Of The Paper:**

This paper presents an extension to the Fourier Neural Operator, called CTFNO that can continuously condition on time, including a universal approximation theorem for the new architecture, and a stability guarantee that should defend against noisy observations. The conditioning on the time is achieved via the adoption of a positional embedding as used in Transformers (Vaswani et al., 2017), and added block wise via shared networks both to the spatial and to the frequency domain.

**Summary Of The Review:**

The ideas are novel and interesting. The current way of how the experiments are presented and compared are weak and hard to judge. I recommend the authors to improve upon the experiments section!

---

> ### Author Response · Authors · 2022-11-08
> **Author response (2) to the reviewer dm3F**
>
> **Q3. In that sense I have also waited for studies which address the effect of the Gershgorin disc normalization..**
>
> **R**:
> Time modulation has been proposed to enable continuous time treatment, not introduced to improve stability.
> Besides, the Gershgorin disc normalization guarantees stability.
> We have already shown in empirical studies that the Gershgorin disc normalization practically enhances the robustness of the model against adversarial attack (Tabs. 5 and 6) and generalization (Fig. 3) in Sec. 4.3. Additional studies have also been included in the Appen. D.2.
> In Fig. 3, we have presented the train and test errors of our model with varying Gershgorin normalization parameter $M$. Similar experimental results on image classification have been reported in Fig. 6.
> The results show that stabilization prevents overfitting and helps the model to work better on unseen data.
>
> **Q4. I am also missing parameter and runtime comparisons.**
>
> **R**:
> The number of parameters of each model has been presented in Appen. C.3 (Tabs. 7, 8, 9, and 10). Also, Tab. 13 in Appen. D.4 summarizes the training and inference time comparisons.
>
> **Q5. I still missing the real intuition why the MuJoCo experiments.. In the current version,..**
>
> **R**:
> In this work, we study PDE solution operators with a focus on real-world problems.
> Following the goal of Neural ODEs (Chen et al. 2018, Rubanova et al. 2019), we would like our model to be applied to various real-world time-series problems by learning the dynamical systems of given data.
> As benchmark examples of real-world problems, experiments on MuJoCo, PhysioNet, and Human Activity have been carried out.
> All the experimental settings and implementation details have been provided in Appen. C in our original submission.
> Containing all of these in the main paper as much as possible would have been ideal, but unfortunately, we provided them in the Appendix due to space limitations.
>
> We are delighted by the reviewer's positive evaluation for the ideas of our work.
> Furthermore, we hope that the above response and the updated draft could address your questions.
> If so, we would like to respectfully ask the reviewer to consider raising your score for this submission.

---

> > ### Comment · Reviewer_dm3F · 2022-11-24
> > **Acknowledging the rebuttal**
> >
> > I'd like to thank the authors for the detailed comments, and explanations. I do see the points related to Q1 and Q2 as important contributions, which were hard to understand (at least for me) when originally reviewing the paper. I have therefore raised my score.

---

> > > ### Author Response · Authors · 2022-11-24
> > > **Thank you very much**
> > >
> > > We would like to thank the reviewer for the reply and raising the score. We highly appreciate it!

---

> ### Author Response · Authors · 2022-11-08
> **Author response (1) to the reviewer dm3F**
>
> We thank the reviewer for valuable comments. We carefully note that many of the reviewer's comments have been primarily included in the Appendix of the original submission, due to the space limitation. We hereby address your concerns as follows:
>
> **Q1. As such a lot of different techniques have been developed..**
>
> **R**:
> Thank the reviewer for the valuable comment. We agree that there may be other methods to impose the temporal information in the network.
> However, we would like to remind t he reviewer that in this work, we focused on **where** to put the time variable, not **how** we put it into the network.
> To maintain the PDE perspective, referring to Green's formula (Eq. (3)), we discussed in Sec. 3.2 that the time variable should be injected into the Fourier weights rather than the input (Eq. (4)). Thus, we used weight modulation, which is a simple and effective way to embed a variable into weights. To verify that such an embedding scheme is effective, we made three baselines (Tab. 1) that integrate time information in an input rather than in weights. As shown in Tab. 1, we empirically verified that embedding time into the weights is much better than embedding it into the input.
> Moreover, as presented in Appen. C.2, we already used sinusoidal embedding for time embedding networks $\phi$ and $\psi$.
> Furthermore, Gupta et al. (2022) embeds the time variable in a similar way to our Baseline 2 in Tab. 3.1. Baseline 2 uses sinusoidal embedding and then concatenate with lifted input. The architectural details of the baseline can be found in Appen. D.1.
>
> **Q2. Long rollouts.. Input/Output for heat equation.. and also for FNO2d.**
>
> **R**:
> Let us denote by $u_t(x) = u(t, x)$ the solution at time $t$.
> Then, the input and output of the original FNOs and ours can be written as follows:
> * FNO1d (autoregressive) : $u_{0} \mapsto u_{\triangle t} \mapsto u_{2\triangle t} \mapsto \cdots \mapsto u_{T\triangle t}$ (for fixed $\triangle t$).
> * FNO2d : $u_{0} \mapsto \left(u_{\triangle t}, u_{2\triangle t}, \ldots, u_{T\triangle t}\right)$ (for fixed $T$, fixed $\triangle t$).
> * CTFNO : $\left(u_{0}, \left(t_1, \ldots, t_T\right)\right) \mapsto \left((u_0, t_1), \ldots, (u_0, t_T)\right) \mapsto \left(u_{t_1}, \ldots, u_{t_T}\right)$ (for arbitrary $T, t_1, \ldots, t_T$).
>
> For example, in the case of heat equation ($T = 50, X = 1024$), the input/output shapes of both FNO2d and CTFNO are $(B, 1, 1024) \mapsto (B, 50, 1024)$ ($B$: batch size, $T$: number of timesteps, $X$: spatial resolution).
>
> We would like to note that our CTFNO predicts the solution at a desirable sequence of time with arbitrary length in one shot, rather than the rollouts.
>
> In Appen. D.5, error heatmaps of reaction, heat, and Burgers' equations for long-time predictions have already been depicted in Figs 8, 9, and 10, respectively.
> Given the initial function, the reaction and heat equations predict solutions for 50 timesteps and Burgers' for 200 timesteps.
> The results show that our CTFNO is able to predict the solution for long timesteps compared to existing methods.

---

### Official Review · Reviewer_FEU2 · 2022-10-20

**Confidence:** 3
**Correctness:** 3
**Technical Novelty And Significance:** 3
**Empirical Novelty And Significance:** 3
**Recommendation:** 5

**Clarity, Quality, Novelty And Reproducibility:**


Clarity: weak
Quality: High
Novelty: Unclear
Reproducibility: poor, I couldn’t code this.


**Strength And Weaknesses:**

s: Fantastic results.

s: perhaps a principled method (maybe, not sure)

w: Weak clarity. I didn't understand the method, and hence I can't evaluate it if has value. The paper follows the convention where equations are given, but not explained, exposed or illustrated. The reader is assumed to be an expert in this domain, which limits the papers audience and thus impact. For instance, eq1 is already weakly presented, but afterwards the computation flow, Nemytskii and FNO are gibberish to me. Similarly eq 4 is just given as a blurp with no explanations. I couldn't follow the time modulations. I didn't understand the ablations: why do we care about this? (The stability stuff was very clear). Furthermore, the model itself is not even defined (!), except in fig2 that I can't follow. In experiments I couldn't follow what task was being solved, and whether table 2 was training or test errors, or what the percentages or pred/interp. settings meant in table 3.

w: Poor motivation. The sec 3.1. explains motivation, but I can’t follow what problem is being solved. Apparently ODEs have some issues fitting toy cases, but then we move to PDEs instead (without citing neural PDE works, or comparing to them, eg. Iakovlev'21). The main motivation is perhaps having a model that can handle both ODE and PDEs, but it’s unclear why do we want this.

w: poor contextualisation. The results are fantastic, but there is little contextualisation on how this method differs from others, or what are its limitations. Does this method solve now everything, does it have any weaknesses?

**Summary Of The Paper:**

A continuous-time version of NFO is presented, which can learn both ODEs and PDEs.


**Summary Of The Review:**

This paper presents an *DE method that achieves amazing results, but does practically nothing to explain either what is happening or what is being learned or why the method is so good. I don’t think just getting SOTA results is science, one also needs to provide insights or the "why".

---

> ### Author Response · Authors · 2022-11-08
> **Author response to the reviewer FEU2**
>
> We thank the reviewer for valuable comments. We hereby carefully address your concerns as follows:
>
> **Q1. Clarification and Motivation**
>
> **R**:
> We sincerely sympathize with the reviewer's point that the fundamental contents could not be fully conveyed.
> Accordingly, we have revised the background section to explain the equations and mathematical formula in a more readable form.
> But, our paper still requires some background on differential equations and FNO. The main motivation and overall context of our paper are as follows.
>
> * **Motivation**:
> Differential equation is a mathematical language that describes the rate of change and interaction of continuously varying quantities of real-world phenomena. After being sparked by Neural ODE, several works have tried to describe the observed time-series data as differential equations. By leveraging PDEs, a much broader range of equations that includes ODEs as a particular class, our goal is to propose a model that can continuously model time-series data across a range of tasks.
>
> * **Architecture of CTFNO**:
> Given an observed initial function, the solution of PDE is mathematically expressed as a convolution of the initial function with the kernel called Green's function (Eq. (1)). FNO approximates the PDE solution operator by learning this Green’s function showing enormous success in various complex PDEs. We extend FNO to a continuous-time aware model to model the time-series data. The solution formula of time-dependent PDEs (Eq. (3)) shows that the weights of FNO layers should be conditioned on time to learn a continuous-time operator. According to this theoretical observation, we integrate the time information into weights of FNO layers via weight modulation (Eq. (4)). Through ablation studies in Tab. 1, we verified the effectiveness of modulating temporal information to weights (Appen. D.1).
> Overall architecture and model design can be found in Appen. C.2.
>
> * **Comparison with other models**:
> Other Neural PDE models we compared are as follows: DeepONet (Lu et al. 2019), POD-DeepONet (Lu et al. 2022), and FNO2d (Li et al. 2021).
> In the case of Iakovlev et al. (2021), which the reviewer mentioned, we think it is another good study that proposed a novel methodology for Neural PDEs.
> Nevertheless, it is very unfortunate that their source codes have not been modified and operated under our experimental settings due to resource limitations.
>
> Intensive experimental evidence confirms that the proposed model outperforms existing methods in a wide range of applications across synthetic and real-applications. We believe that this work proposes a novel framework of learning a broad range of real-world time-series problems.
>
> **Q2. Experiments**
>
> **R**:
> All the experimental settings for Sec. 4.2 have been described in Appendices C.1, C.3, which are the same as those of our benchmark, Latent ODE-RNN (Rubanova et al. 2019).
> The values in Tab. 2 refer to test errors, not training.
> The percentages in Tab. 3 indicate how much input timesteps are exposed to the models.
> For interpolation, if the original input has a total of $100$ timesteps and exposes $30\%$, we mask the remaining $70$ timesteps randomly and show it to the model, and the model aims to restore the total of $100$ timesteps.
> For prediction, if the original input has a total of $200$ timesteps, the first half of $100$ is used as the input after masking by the ratio specified in the Tables, and the goal is to match the second half of $100$ (In case of MuJoCo, $2/3$ of timesteps should be predicted).
> Containing all of these in the main text as much as possible would have been ideal, but we are sorry that it was difficult because of the page limitation.
>
> **Q3. Limitations**
>
> **R**:
> As written in conclusion (Sec. 6), one of the limitations of CTFNO is that it is difficult to embody discontinuous features due to the limitation of the Fourier transform that only captures global information.
> Another limitation is that good performance could not be guaranteed for time-series data which cannot be described by PDEs.
> This is also the reason why the scope of the problem was focused solely on dynamics-based time-series data.
>
> Finally, we hope this response has contributed to the empathy of the problem consciousness and a broad understanding of CTFNO.
> If there is any beneficial improvement in clarity, motivation, and contextualization through this reply, we would like to kindly ask the reviewer to consider raising the score.

---

> > ### Comment · Reviewer_FEU2 · 2022-11-10
> > **Discussion**
> >
> > Thanks for response. I checked the paper changes, and they do little to address my concerns. I think this is great work with great contribution, but the paper does not do the work justice yet. For ICLR it should to be accessible to wider ML audience.
> >
> > My standing issues are:
> > * The mathematical treatment is incomplete. There are many (fully or partially) undefined/unexplainedr terms (eg. x, y, G, PDE, time derivatives, DE solution, \cal{K}, phi, F, the model itself, loss, regularisation, RNN encoder, P/Q, etc..). The technical presentation needs to be precise and self-contained, at least for the main parts of the story.
> > *  There is not enough exposition to follow the story at conceptual level. Things are given with no explanation. The main beef of the paper is the temporal frequency modulation. I would then expect the paper to introduce and expose this approach in a manner that the reader can follow (eg. frequency visualisations could help).
> > * The results give no insight into what the system learns, or how/why does it improve. Having just benchmark tables gives me very little. One could eg. study the learnt frequencies or maps, or do ablations. The system seems to have massive capacity to fit trajectories perfectly, but why don't we then overfit? For instance, a very insightful experiment would be to continue the spiral forecasting of fig1 for couple more loops (without adding more training data): does the model converge to (0,0)?

---

> > > ### Author Response · Authors · 2022-11-13
> > > **Thank you for feedback**
> > >
> > > We sincerely thank the reviewer's further valuable suggestion and comments. Please find our responses below. Revisions have also been made in the updated manuscript and all corrections are marked in blue color.
> > >
> > > **Q.1 Incomplete mathematical and technical representations.**
> > >
> > > **Reply**: We agree that some notations and implementation details are unclear or have been used without defined. Thank you for the reviewer's careful reading and pointing this out.
> > > We have clarified mathematical notations (especially in Secs. 2, 3.1, and 3.2) in the updated manuscript.
> > > The training losses of PhysioNet, Activity, and classification have already been provided in Appens. C.3.2 and C.3.3.
> > > We have included training loss in Sec. 4.1 for synthetic, Appen. C.3.2 for MuJoCo, and Appen. C.3.3 for Plane Vibration experiments in the updated version.
> > > A detailed description of the RNN encoder has been added to the Appen. C.3.2 (a paragraph  'Latent ODE framework'). We have also added in Sec. 2 that $P$ and $Q$ are linear.
> > > Furthermore, we rephrase the motivation section (Sec. 3.1) to address your concern on unclear motivation behind the proposed PDE-based approach for modeling time-series data. We have also added a brief introduction of PDEs in Sec. 3.1 in the revised manuscript.
> > > We hope we had fully addressed the clarification issue.
> > >
> > > **Q.2. Unclear explanation of time modulation.**
> > >
> > > **Reply**: One of the reviewer's main concern is unclear description of the proposed time modulating scheme.
> > > To address the reviewer's concern, we made several changes that improve the manuscript.
> > > Our proposed time-dependent Fourier layer has been defined in Eq. (4).
> > > But, we have included more definitions (Def. 2.1,  3.1) to improve the readability in the revised paper.
> > > How Eq. (4) is parameterized has also already been specified in the 'Time Modulating' paragraph in Sec. 3.2;
> > > two sharing networks consist of two-layer fully connected networks after sinusoidal embedding of $t$ (the contextual visualization of sharing net also can be found in Fig. 4),
> > > time modulation operators $A$ and $B$ are affine transforms.
> > > The parameterization of $R$ and $W$ have been contained in Sec. 2, but to make it clearer, it have been mentioned again that $R$ and $W$ are parameterized by complex and real matrices, respectively, in the revised paper. We refer the reviewer to Appen. C.2 for more details. We have further added algorithms of our CTFNO in Appen. C for the purpose of better understanding.
> > >
> > > **Q.3. No insight from experiments and benchmark tables.**
> > >
> > > **Reply**: It is difficult to agree with the reviewer's concern that the benchmark table implies only simple or weak results.
> > > Interpolation and prediction tasks are not the first to be presented in this paper, but they are still challenging experiments performed by many works, including Latent ODE-RNN and Neural Flow.
> > > The fact that CTFNO achieved state-of-the-art performance proves that it can be significantly used in the real-world applications.
> > >
> > > Furthermore, in addition to Tabs. 3 and 4 for comparison with benchmarks on MuJoCo and PhysioNet, we conducted various experiments and reported results to support the argument.
> > > For instance, the results of our ablation study tabulated in Tab. 1 assert that the proposed time modulation method, which is theoretically grounded, outperforms other time-equipping methods.
> > > Tab. 2 implies that CTFNO can handle not only PDEs but also ODEs, even curves with cusps, which is distinctly different from other models. (These are verification experiments for our argument in the motivation section.)
> > > Fig. 3, Tab. 5, and Tab. 6 demonstrate the effect of stabilization method we proposed, and one can verify the robustness through these ablation studies.
> > >
> > > **Q.4. Generalization capacity of the model.**
> > >
> > > **Reply**: We appreciate the reviewer for the thoughtful suggestion.
> > > Following the suggestion, we performed additional extrapolation experiments on spiral data. We trained each model (our CTFNO and baselines) to predict spiral trajectories at 100 time points, and then test the trained models to extrapolate to 500 future time points. We observe that our model is better at generalizing for extrapolation than baselines. We have added these experimental results in Fig. 8 in Appen. D.5. Judging from these results, it can be seen that our model correctly captures the dynamics of the spiral during training, not simply memorizing it. Therefore, it achieves superior performance in both prediction (the lowest RMSE in Tab.2) and extrapolation (Fig. 8).
> > >
> > > Again, we'd like to warmly thank the reviewer for your time and thoughtful feedback which helps to improve the quality of our work. We also believe that the reviewer's suggestions have led us to further improve the clarity of the manuscript as well as to add more technical details.
> > > We hope we had fully addressed the reviewer's comments and concerns. We do hope these responses and the revised manuscript are helpful for the reviewer to re-evaluate our paper.

---

> > > > ### Comment · Reviewer_FEU2 · 2022-11-14
> > > > **starting to understand the method a bit**
> > > >
> > > >
> > > > There are some mistakes
> > > >
> > > > * eq1 has u of size d_u, and \cal{G} of size d_a. Same mistake happens in eq 3, where G and a are d_a.
> > > > * I’m confused by def 2.1. \cal{K} is linear operator. How is it defined? Is it again a matrix product similar to P/Q? What is “fourier transformed kernel”? Is this a fourier transform of a kernel perhaps? No kernel has been defined, but perhaps this points to \cal{K} being a kernel. However, are kernels linear? What do you mean by kernel (there are alternative definitions)? What does it mean to take a fourier transform of a linear operator: does this give us something useful? Intuitively it’s difficult to see that Fourier of a matrix would make much sense. Finally, if R is F(\cal{K}) (is it?), then will it have imaginary components if \cal{K} is linear? I’m also confused how do you compute the fourier transform in practise, over what? The \cdot is confusing: it looks like inner product, but hasn’t been defined. The dimensions don’t seem to make sense for an inner product.
> > > > * Where is the frequency \xi coming from in def 2.1.? The v and x are clearly inputs that we choose, but to evaluate them one needs to know one \xi (which one?). Shouldn’t you look at all frequencies? Is there an integral hidden somewhere here? Perhaps the \cdot hides this, but shouldn’t you in that case drop the indexing variable \xi? Same thing happens in eq 4: you need to know one magic frequency for some reason.
> > > > * It would be good to define the Green function in eq 3 (even informally, for instance, “there exists a function G such that u(x,t) =..")
> > > > * Overall I don’t think sec 2 mathematical formulations are yet complete.
> > > >
> > > > I think I start to understand what the paper is doing: You are looking at the frequencies of the current function surface $v$, and multiplying them by the frequencies of some ambient data-space kernel, with a linear temporal scaling component. You are then adding this contribution with an ambient perceptron part. Why do we need both?
> > > >
> > > > Furthermore, how do you compute the F(v)? I don’t see how can you know the entire function v to do this since it’s only being generated layerwise at a single point (t,x) only. Similarly, to know R you need to know the kernel it was computed from. I think the kernel was said to come from G, but then how do you know G? Surely each PDE has a different G? Are you actually assuming knowledge of G? Please define what is R and G and the “kernel” that is being referred to (\cal{K}?), and whether its linear or not.
> > > >
> > > > In your response you say that R and W are “parameterised” by matrices. But R was said to be a fourier transform in the main text! But then in "time modulation" you start to refer R as "fourier kernel" which sounds like just a matrix (ie. not a kernel). I guess so, but how do you get this then? Can you define what you mean by "kernel" in general?

---

> > > > > ### Author Response · Authors · 2022-11-15
> > > > > **Thank you for comments!**
> > > > >
> > > > > We warmly thank the reviewer for your quick reply and detailed review and constructive comments.
> > > > > We’ve updated  the revised version of our paper (with track changes marked in blue).
> > > > > Our response to your comments are listed below:
> > > > >
> > > > > **Q1. Issues about $G$, linearity of $\mathcal{K}$, notion of kernel, mathematical formulations in Sec.2.**
> > > > >
> > > > > **Reply**: We would like to apologize for our incorrect statement of the codomain of $G$. It is of size $d_u\times d_a$. Thank you for pointing this out. Also, to be precisely, $\mathcal{K}$ is a convolution operator, not a linear operator. As the reviewer pointed out, another confusion seems to be caused by the use of the term 'kernel' without defining it clearly. For integral operators such as convolution operators, a constant object around which input functions are integrated, like $G$ in Eq. (1), refers to the kernel.
> > > > > We are deeply sorry for the confusion caused by our mistakes. We have clarified all of these in the revised manuscript.
> > > > > Moreover, it seems that our misstatement of $\mathcal{K}$ as a linear operator hinders the reviewer from understanding the Fourier layer defined in Def. 2.1.
> > > > > But in fact, Fourier layers are designed based on the useful fact that the Fourier transform of a convolution of a function with the kernel is the product of their Fourier transforms.
> > > > > In Def. 2.1, $\mathcal{K}$ is a convolution operator with kernel $\kappa$
> > > > > and $R$ is the Fourier transform of the kernel $\kappa$.
> > > > > According to the definition, $R$ is clearly a complex matrix.
> > > > > We carefully checked the mathematical formulations in Secs. 2, 3.1, and 3.2 to make sure there were no more incorrect or unclear terms. We hope that the updated manuscript is clear to you as well.
> > > > >
> > > > > **Q2. Issues about frequency $\xi$ in Eqs. (1) and (4). How to compute the Fourier transform $\mathcal{F}(v)$?**
> > > > >
> > > > > **Reply**: We are grateful for the reviewer's valuable comments.
> > > > > Our intention was to describe the fact that the input function of inverse Fourier transform is in the frequency domain.
> > > > > Nevertheless, we agree that the expression of frequency $\xi$ in Def. 2.1 may cause confusion to the readers.
> > > > > Hence, in the revised version, we removed $\xi$ from Eqs. (1) and (4), and described what the operation $R_{\ell} \cdot \mathcal{F}(v)$ on the frequency domain denotes, instead.
> > > > > We also hope that it is now clear what the symbol \textbackslash cdot refers to, in Def. 2.1. (As a result, it's a natural matrix multiplication.)
> > > > >
> > > > > Meanwhile, in practice, we regard $x$ as a fixed grid to recognize $v(x)$.
> > > > > This is because all the values of $v(x)$ on $x \in \Omega \subseteq \mathbb{R}^{n}$ can be uncountably infinite and thus computationally intractable.
> > > > > Note that the size of the grid or the length of each interval can be determined arbitrarily.
> > > > > For instance, the original FNO (Li et al. 2021) has set the grid as an equidistant interval (i.e. $\left[0, 1/n, \ldots, (n-1)/n \right]$).
> > > > > Thus, $v(x) \in \mathbb{R}^{d_v}$ is a $d_v$-dimensional vector of which length is the same as the $x$-grid size.
> > > > > To compute $\mathcal{F}(v)$, we can now apply the well-known fast Fourier Transform (FFT) algorithm to $v(x)$ since it is discrete and finite.
> > > > > One may concern about the assumption of such grid, however, the resultant solution $u(x)$ can be meaningful if the resolution (i.e. the grid size) is sufficiently high.
> > > > >
> > > > > **Q3. It would be good to define the Green function in eq 3.**
> > > > >
> > > > > **Reply**: Thank you for the suggestion. We have revised the corresponding sentence as you suggested.
> > > > >
> > > > > **Q4. Why did you add the time-embedded kernel term to the ambient perceptron part together? Why do you need both?**
> > > > >
> > > > > **Reply**: The description of $W$ and $R$ in Def. 2.1 originated from the FNO (Li et al. 2021). The authors of FNO showed that both $W$ and $R$ have their own roles, and they explained the necessity of both terms. We think original FNO will solve part of your questions, so please refer to FNO.
> > > > > Extending the previous work, we injected time information into both of $W$ and $R$.
> > > > > Moreover, to guarantee the universality of CTFNO (Thm. 3.1, Appen. A), the time $t$ should also be injected to $W$.
> > > > > For these reasons, we put $t$ in both terms and add them together.
> > > > >
> > > > > **Q5. I think the kernel was said to come from G, but then how do you know G? Are you assuming knowledge of G?**
> > > > >
> > > > > **Reply**: Under fairly general conditions on PDEs, it is well known that the solution to the give PDE is expressed as a convolution with $G$, which is called the Green's function. The purpose of FNO is to approximate PDE by recovering $G$ through network learning. In other words, the goal is not to assume that you know $G$, but to learn $G$ from the data (input and output). $G$, $R$ and $\mathcal{K}$ are now clearly redefined in revised manuscript (Section 3.2).
> > > > >
> > > > > We hope our responses and the updated manuscript had fully addressed the reviewer's concerns and comments. Please do feel free to let us know if you have any further questions. Thank you very much.

---

> > > > > > ### Comment · Reviewer_FEU2 · 2022-11-24
> > > > > > **response**
> > > > > >
> > > > > > Thanks for the responses.
> > > > > >
> > > > > > I believe the paper is more or less well-defined now in terms of notation, while the model is still not fully defined (eg. loss is missing). However, this then raises a next issue, which is weak'ish clarity. There is lots of notation to follow, and its not clear what intuitively lots of this stuff means or models, or why do we need them. The problems with result exposition stand as well.

---

> > > > > > > ### Author Response · Authors · 2022-11-25
> > > > > > > **Response**
> > > > > > >
> > > > > > > We thank to the reviewer for reply.
> > > > > > >
> > > > > > > We have defined our model rigorously in Sec. 3.2 and improved clarity by adding the explicit algorithm of our Fourier layers (Algs. 1, 2, and 3) to Appen. C.
> > > > > > > Moreover, we have also stated the loss function as follows:
> > > > > > >
> > > > > > > #### **[Synthetic data]**
> > > > > > > * For all experiments, we used MSE loss.
> > > > > > >
> > > > > > > Please see Sec. 4.1 (bottom of page 5).
> > > > > > >
> > > > > > > #### **[Real time-series]**
> > > > > > > * For interpolation and prediction, we used the importance weighted likelihood loss (Burda et al., 2015).
> > > > > > > * For classification (Activity), we used the cross entropy loss.
> > > > > > >
> > > > > > > Please see Sec. 4.2 (top of page 7) and Appen. C.3.2 (bottom of page 24).
> > > > > > >
> > > > > > > #### **[Stability]**
> > > > > > > * For Plane Vibration, we used MSE loss.
> > > > > > > * For image classification, we used the cross entropy loss.
> > > > > > >
> > > > > > > Please see Appen. C.3.3 (top of page 25).
> > > > > > >
> > > > > > > We believe that discussions with the reviewer have made our paper more rigorous and enhanced its clarity.
> > > > > > > We hope our revised manuscript will have a positive effect on the reviewer's overall evaluation.
> > > > > > > Thank you for your valuable comments.

---

### Official Review · Reviewer_Jaqe · 2022-10-23

**Confidence:** 5
**Correctness:** 3
**Technical Novelty And Significance:** 3
**Empirical Novelty And Significance:** 3
**Recommendation:** 5

**Clarity, Quality, Novelty And Reproducibility:**

There are a few typos in the paper. Core idea of conditioning neural-net layers with time is fairly commonly done in diffusion models, transformers, NeRFs etc. these days. However how to do it w.r.t. Fourier layers is new here. Provided code is easy to follow and the results should be easy to reproduce.

**Strength And Weaknesses:**

**Strengths**

Time-conditioning in the Fourier space as done in this paper is definitely novel. Similarly row-wise normalizing the Fourier kernel is a useful trick to know. The paper applies their method on a very wide variety of problems, demonstrating the general applicability of their approach.


**Weaknesses**

Claims abut lack of expressiveness of ODEs in the second introductory paragraph are incorrect. NeuralODEs are difficult to train in practice on higher dimension problems and there have been many different approaches to try fix some of these issues in various individual contexts. S4 [1,2] class of methods actually try to solve this problem generally while being computationally less expensive. Proposing a "PDE" based framework seems ill-motivated and calling FNO as a PDE solver is also incorrect. A PDE solver will solve the given equation at whatever timestep you want answers for, FNOs first need to be trained on solutions from that PDE solver to make a "PDE surrogate" and can still only "solve" that particular equation and will not generalize to change in the parameter of that equation. Similarly as this paper notices too that FNO has a notion of discrete time and is often trained to recursively make predictions and can't give you predictions at whatever timestep you want. I know the literature in this space is a mess and I don't blame the authors per se but some precision in our language is required for effectively communicating with the differential equations community if you are supposedly solving problems with their approaches.

Mention of U-net based diffusion models is another remiss. There too scalar time is used to condition the different layers of neural network in a manner that is not _too_ different from what this paper proposes. [3] demonstrates how applying the same techniques and architecture is quite effective for PDE modeling (orders of magnitude better than FNO) and also does time-conditioning similar to this paper.
Overall, expressiveness of FNOs is the reason for superior performance and more details about training strategies and baselines is necessary to actually make sense of results.

In the Image classification experiments it's unclear what CNN model was used as baseline and there are methods to improve CNN style model accuracies too which would ideally be compared as well. I'm not sure if it even should be in main text of the paper, it just seems so disjoint from the title. Moreover it's unclear if this "stabilization" won't hurt performance on difficult 2D PDE modeling tasks like Navier-Stokes.

[1] https://arxiv.org/abs/2111.00396

[2] https://arxiv.org/abs/2206.03398

[3] https://arxiv.org/abs/2209.15616

**Summary Of The Paper:**

Similar to as is done in done in diffusion models for U-net layers, this paper applies time-conditioning to FNO layers and demonstrates it applicability on a wide variety of tasks. The paper also proposes a method to normalize the Fourier kernel so that it's more perturbation resistant.

**Summary Of The Review:**

Overall there are useful ideas in the paper. However it needs better focus on its identified contributions. Clear discussion of limitations would also be quite useful.

---

> ### Author Response · Authors · 2022-11-08
> **Author response to the reviewer Jaqe**
>
> Thank you for providing detailed feedback and suggestions. We hereby carefully address your concerns as follows:
>
> **Q1. Claims about lack of expressiveness of ODEs.. Calling FNO as a PDE solver is also incorrect ..**
>
> **R**:
> Thanks for pointing out our argument about lack of expressiveness of ODEs. We rephrase this part following the reviewer's advice.
> We also agree with the reviewer's concern that it is hard to refer FNO by a PDE solver. However, because FNO is a neural network that tries to approximate the solution operator of PDEs and is commonly regarded as a neural PDE surrogate, we would like to continue the argument that our methodology is a PDE-based framework.
> Based on the reviewer's comment, the term 'PDE solver' is no longer used in our paper and we have updated the corrected manuscript.
>
> **Q2. Mention of U-net based diffusion models is another remiss.. Overall, the expressiveness of FNOs is the reason for superior performance..**
>
> **R**:
> We would like to begin by pointing that the expressive power of FNO is not a main factor of the performance of our CTFNO. If so, the performance difference between the three baseline models of the ablation study (Tab. 1) and our CTFNO should not have been that much. This observation shows that CTFNO has a structure that is more suitable to handle time in addition to the expressive power of FNO.
> As the reviewer has pointed out, there are many different ways to put time into the network. In this work, we tried to put time information into the weights of the network based on Green’s function formula (what we want to approximate). The ablation study (Tab. 1) further demonstrated the effectiveness of our time modulation than putting time information to the input/feature vectors such as diffusion models or Transformer did. Since the main discussion of our work is where to put time information in the network, we did not include the comparison with other models suggested by the reviewer (such as diffusion models, Transformer). Also, the time embedding method of Gupta et al. (2022) is similar to Baseline 2 in our ablation study, but a comparison cannot be carried out because an open-source code is unavailable.
>
> **Q3. In the Image classification experiments it's unclear what ..**
>
> **R**:
> Our main purpose of the classification experiment is to verify whether our proposed stabilization is effective and applicable.
> As part of it, we evaluated the robustness of the classifiers against adversarial attacks.
> Since the new stabilization technique is one of the main contributions, we still think the robustness experiments should be regarded as one of the main results.
> For a fair comparison, we set the parameters of the networks to similar levels; a simple CNN, ANODE, original FNO, and the weight regularized FNO.
> The network architectures have been provided in Appen. C.3.3.
>
> **Q4. Does the stabilization hurt performance on difficult 2D PDE modeling tasks like Navier-Stokes?**
>
> **R**:
> We thank the reviewer for the valuable comment.
> When we approximate a solution to an ill-posed PDE (in the sense of Hadamard), our stabilization can degrade the performance of the model.
> In this case, in order not to impair the performance, we could train the model with a large stabilization parameter $M$.
> However, $M$ is a user-defined hyperparameter, and thus we still do not know the optimal value for $M$.
> We have added this limitation in Sec. 6 in the revised manuscript.
>
> We hope that our response can address all concerns raised by the reviewer. If so, we would appreciate if the reviewer reconsider the novelty and contribution of our work.

---

### Official Review · Reviewer_92q6 · 2022-10-27

**Confidence:** 3
**Correctness:** 4
**Technical Novelty And Significance:** 2
**Empirical Novelty And Significance:** 3
**Recommendation:** 6

**Clarity, Quality, Novelty And Reproducibility:**

The means and standard deviations of results in the paper should be provided.
There are some typos but can be improved.


**Strength And Weaknesses:**

### Strength
- The paper is well-written, and well-structured.
- Theoretical results support the claim on the stability of the proposed method.
- Experiments are comprehensive, covering multiple aspects from learning dynamics and robustness to adversarial attack.
### Weaknesses
- Although the suggested time modulation shows impressive performance, it’s hard to justify why it works well with this specific choice (in Table 1). I’m not familiar with StyleGAN2. So if there is any good justification from StyleGAN2, it would be nice to have a brief discussion of the benefits of such an architecture design.


**Summary Of The Paper:**

The paper presents an extension of Fourier neural operators (FNO) to have operators evolving over time. The paper proposed an architecture to cooperate the time embedding into FNO. The experiments are conducted on synthetic datasets given PDEs and on real-world time series data. The model outperforms alternative methods. There is also an experiment on the robustness to adversarial attacks.

**Summary Of The Review:**

The paper did a good job describing the proposed method and the emperical results are impressive. However, it's not clear what factor leads to the effectiveness of the proposed models.

---

> ### Author Response · Authors · 2022-11-08
> **Author response to the reviewer 92q6**
>
> We thank the reviewer for the positive feedback and thoughtful comments. We hereby carefully address your concerns as follows:
>
> First of all, we would like to emphasize that the effectiveness of our model mainly comes from where we put the time in the network, not from the modulation technique inspired by StyleGAN2.
> The primary goal of our work is to model real time-series through a continuous-time PDE solution operator.
> Starting from a time-dependent Green's function formula Eq. (3), we propose a newly designed time-aware Fourier layer by inserting temporal information into weights (as in Eq. (4)).
> Moreover, ablation studies in Sec. 3.2 verified that adding temporal information to weights is much more effective for learning PDE solution operator than integrating them into input/output tensors (See Tab. 1).
>
> We also want to clarify that we do not use the weight modulation of StyleGAN2 as it is.
> Due to the architecture of FNO, we need two kinds of time representation vectors acting on the Fourier domain $\phi(t)$ and the spatial domain $\psi(t)$, and both are attained via a popular sinusoidal embedding.
> As is well known, there are several methods to combine time information in neural networks.
> In particular, we would like to emphasize that $\phi(t)$ and $\psi(t)$ should be combined with the weight tensor of each layer, not with an input or output tensor, which implies the necessity of 'weight modulation.'
> One suitable option is the novel weight modulation method proposed in StyleGAN2.
>
> Briefly, StyleGAN2 generates various images by reflecting a 'style vector' (gender, skin color, hair color, etc.) on each convolutional weight.
> Their methods contain modulation, scaling the convolution weights by a representation vector, and demodulation, normalizing each output feature by the $L^{2}$ norm of the corresponding weights.
> To make it more suitable to our model structure which learns the kernel in the Fourier domain, we modified StyleGAN's method as follows:
>
> * Modulation: Motivated by StyleGAN2, we scale the weights based on $\phi(t)$ and $\psi(t)$, per-frequency in the Fourier domain (upper line in Fig. 2) and per-channel in the spatial domain (lower line in Fig. 2), respectively.
> We adopted this because it was an intuitive and effective way to modulate time representation vectors to weights.
> Nevertheless, there is a key difference between ours and StyleGAN2: unlike StyleGAN2, we applied modulation to the Fourier and the spatial domains along different axes.
>
> * Demodulation: We did not apply demodulation to our work.
> The main reason is the existence of our proposed Gershgorin discs normalization (Sec. 3.4).
> This stabilization method is very useful for constraining the weights as well as ensuring the stability of the learned PDE.
> According to the Prop. 3.2, the norm of weights should be limited to less than a specific value (like our method), but the demodulation of StyleGAN2 is far from what we want because it normalizes the norm of the weight to a target value.
> Thus, CTFNO did not require a role of demodulation, and this is another difference between ours and StyleGAN2.
>
> As far as we know, this paper is the first to propose a neural operator with temporal information using modulation and achieve successful results on real time-series applications.
> We would like to reiterate that it is important to put temporal information in the weights, according to the time-dependent Green's function (what we want to learn).
>
> We have also updated several typos in the revised manuscript.
> Finally, if there is any beneficial improvement in clarity, motivation, and contextualization through this reply, we would like to kindly ask the reviewer to consider raising the score.

---

### Author Response · Authors · 2022-11-08
**Author Response to All Reviewers**

We thank all the reviewers for their thoughtful comments and suggestions on our paper. We think that the reviewers raised several insightful questions and we believe that answering those questions has significantly improved our work.  We read the reviewers' comments carefully and have updated the paper to incorporate reviewers' advice and further improve the writing and all changes are marked in red color. The corrections for the second response to the Reviewer FEU2 are marked in blue color.
Below, we have responded to the reviewers’ comments in separate replies.

We hope our response and the updated manuscript could address all your concerns. We deeply thank again all the reviewers for their kind and constructive comments.

---

### Author Response · Authors · 2022-11-28
**Reminder**

Dear reviewers,

we would like to kindly remind you that the period of discussion stage 1 is over,
and the period of the stage 2 discussion ends in two weeks.
We would appreciate it if the reviewers could take a look at our responses, and please let us know if our replies address all your questions/concerns
or if you have further follow-up questions.

We appreciate all reviewers for their valuable comments.
Also, we deeply thank the reviewers for their devoted time, energy, and effort.

Thank you very much.


$\ $

Sincerely,

the authors of the paper "Learning PDE Solution Operator for Continuous Modeling of Time-Series"

---

### Decision · Program_Chairs · 2023-01-20

**Decision:**

Reject

**Justification For Why Not Higher Score:**

The main reason is that the paper is not well-written. It uses very informal (even annoying language): i.e. writing a general operator and writing the solution as convolution with a shift-invariant, time dependent function does not help. Why shift-invariance (none of the example have this property). Why not general G(t, x, y)? The main efficiency is probably in the architecture design, but it is not clear when it will work. A simple example is needed (maybe even a simple time series).

**Justification For Why Not Lower Score:**

N/A

**Metareview: Summary, Strengths And Weaknesses:**

The paper proposes a new neural architecture (called continuous-in-time Fourier Neural Operator, CTFNO), which is based on the idea that the Green function for a (linear PDE) should depend on time. Numerical examples are shown.

Strengths: Time conditioning network in the FNO is novel, as well as normalization trick by the Gerschgorin circles. The numerical experiments are quite promising.

Weaknesses: The paper is not well-written and is difficult to follow. As pointed by one of the reviewers, it is not authors fault: it is a mess on the FNO literature nowdays. I.e., the standard FNO approach is not even a mapping from a function to a function (see Fanaskov, Oseledets, "Spectral Neural Operators": the non-linearity in the physical space introduces aliasing and additional harmonics),
so the theorems on the universality of the approach are very questionable.
Second, the sloppiness of the method: the PDE mentioned is written in the general form, whereas the Green function is written with the shift-invariant case, and only holds for the linear operators $\mathcal{L}$ (since the layer is linear). So, the whole motivational part should be presented in a different way in my opinion.

The main missing thing in the submission is a simple example that illustrates the case where the time-dependent method will work (significantly) better. Maybe a simple time-series, but the current informal style of writing, together with a specific, quite complicated time modulation architecture, does not make me believe in the new approach.